# Recursive Monte–Carlo Tree Search

Benjamin Howard [* 1]   Keith Frankston [* 1]

## Abstract

We introduce a recursive AlphaZero-style Monte–Carlo tree search algorithm "RMCTS". It first generates the search tree using prior policies, and then recursively re-estimates action values by using the regularized optimal posterior policies from "Monte–Carlo tree search as regularized policy optimization" (Grill et al., 2020) at each node of the search tree, starting from the leaves and working back up to the root. We find that RMCTS matches or exceeds the quality of AlphaZero's MCTS-UCB in a tiny fraction of the time.

## 1. Introduction

We introduce Recursive Monte–Carlo Tree Search (RMCTS), a drop-in replacement for AlphaZero-style Monte–Carlo tree search. The main idea of RMCTS is to do regularized policy optimization (as in (Grill et al., 2020)) at every node in the search tree, setting estimated action values to be the expected rewards following the posterior policy. By nature of this design, it is possible to explore the search tree in a breadth-first manner, thus massively parallelizing the neural network inferences at each level of the tree. Hence RMCTS is significantly faster[1] than AlphaZero's MCTS-UCB (Silver et al., 2018).

The recursion in RMCTS is based on computing posterior policies at each game state in the search tree, starting from the leaves and working back up to the root. Here we present results using RMCTS with the posterior policy explored in "Monte–Carlo tree search as regularized policy optimization" (Grill et al., 2020). Their posterior policy is the unique policy at a state which maximizes the expected reward given estimated action rewards minus a penalty for diverging from the prior policy.

The tree explored by RMCTS is not chosen in an adaptive manner, as it is in MCTS-UCB. Instead, it is explored according to the prior network policies at each node. Despite this, we find that in head-to-head contests between RMCTS and MCTS-UCB, each using the same neural network for priors, RMCTS needs only a tiny fraction of the time awarded to MCTS-UCB to beat it (see section 7).

When training, since we are typically searching a large batch of independent game states, the advantage of RMCTS over MCTS-UCB is less pronounced, but still significant. Our experiments have shown that RMCTS-trained networks match the quality of MCTS-UCB-trained networks in roughly one-third of the training time, across the three games we studied (see Appendix E).

### 1.1. Related work

To mitigate the latency of sequential search, various parallelization strategies have been explored, including classical tree parallelization using virtual loss (Chaslot et al., 2008). While virtual loss allows multiple threads to traverse the search tree simultaneously by temporarily penalizing nodes under evaluation, its benefits are fundamentally limited by search overhead and strength saturation (Chaslot, 2010). In traditional CPU-bound settings, virtual loss can achieve near-linear playout speedups for modest numbers of threads, but decision quality often plateaus as the search diverges into suboptimal regions to maintain thread independence (Segal, 2010). Further analysis of search overhead using virtual loss can be found in (Mirsoleimani et al., 2017). These issues mainly arise due to the ad-hoc nature of virtual loss. Since RMCTS is designed from first principles, it avoids these pitfalls entirely (cf. section 7).

Modern high-performance libraries like mctx (DeepMind et al., 2021) primarily achieve efficiency through **inter-tree batching**, parallelizing across thousands of independent game environments. However, for a single search instance, standard parallelization remains bound by the sequential selection-inference-backpropagation cycle due to the lack of batching within a single tree. In contrast, RMCTS bypasses this bottleneck and enables **intra-tree batching**, achieving speedups of over $40\times$ by processing each level of the search tree as a single batch. Rather than attempting to parallelize the inherently sequential MCTS-UCB algorithm using ad-

---

[*]Equal contribution   [1]Institute for Defense Analyses, Center for Communications Research, Princeton, NJ, USA. Correspondence to: Benjamin Howard <bhoward73@gmail.com>, Keith Frankston <k.frankston@fastmail.com>.

*Proceedings of the $43^{rd}$ International Conference on Machine Learning*, Seoul, South Korea. PMLR 306, 2026. Copyright 2026 by the author(s).

[1]An efficient implementation is available at https://github.com/bhoward73/rmcts

hoc techniques like virtual loss, RMCTS is designed from the ground up as a principled method for policy improvement, and the parallelizability is a natural feature.

## 1.2. Outline

In section 2, we review AlphaZero's MCTS-UCB, and mention some basic properties of UCB values. In section 3, we review the optimized posterior policy of (Grill et al., 2020), and give an interpretation of this policy in terms of UCB values. In section 4, we introduce RMCTS, and give a precise description of the algorithm. In section 5, we illustrate RMCTS with a simple example. We include timing (section 6) and quality (section 7) comparisons of RMCTS vs. MCTS-UCB for three games: Connect–4, Dots-and-Boxes, and Othello. In section 8, we discuss possible future directions and highlight potential generalizations.

**Conflict of Interest Disclosure**  The authors declare that this research was supported by and conducted during their employment at the Institute for Defense Analyses. The authors have no competing financial interests, corporate sponsorships, or personal relationships that could be construed to influence the algorithmic design or empirical results reported in this paper.

## 2. AlphaZero's MCTS-UCB

AlphaZero (Silver et al., 2018) and its predecessor AlphaGo (Silver et al., 2016) are methods to train neural networks to play games at a high level. The main idea is to modify Monte–Carlo tree search ((Coulom, 2006; Kocsis & Szepesvári, 2006)) so that simulations/playouts at leaf nodes are replaced with values from a neural network. Furthermore, a prior policy from the network is used to guide the search, establishing MCTS-UCB as a combination of policy and value functions. The result of the search is a new (posterior) policy for the root state.

AlphaZero's MCTS-UCB works roughly as follows (see Algorithm 4 in Appendix A for a more precise description). We initialize estimated action values $Q(s, \cdot)$ to zero, and then perform a number of simulations to refine the $Q$ values and obtain a posterior policy. A simulation always begins at the root state, and chooses an action with the maximal UCB value, where the UCB value is defined to be

$$\text{UCB}(s, a) = Q(s, a) + C \cdot \pi_0(s, a) \cdot \frac{\sqrt{\sum_b N(s, b)}}{1 + N(s, a)}$$

where $s$ is the current state, $a$ is an available action from $s$, $Q(s, a)$ is the estimated reward for taking action $a$ from state $s$, $C > 0$ is the exploration constant, $N(s, a)$ is the number of times action $a$ has been selected from state $s$ in previous simulations, and finally, $\pi_0(s, \cdot)$ is the prior (network) policy.

We continue picking the action with highest UCB, moving down the exploration tree, until we reach a state $s'$ that is terminal, or which is not yet in the tree. If $s'$ is terminal, then its value is simply the game score. Otherwise, the value of $s'$ is defined to be the network value $v_0(s)$, and $s'$ is added to the tree, initializing $Q(s', \cdot)$ and $N(s', \cdot)$ to zero for all actions from $s'$. The value of this final state is propagated back up the path to the root using the appropriate sign for the active player; each state along this path updates its $Q$ and $N$ values appropriately.

Updating the $Q$ and $N$ values causes the UCB values to change, and so a different path may be taken in the next simulation. Once $N(s) = \sum_a N(s, a)$ reaches the budgeted number of simulations, the process stops. At this point, the posterior policy $\hat{\pi}$ at a state $s$ is defined to be the normalized visit counts $\hat{\pi}(s, a) = \frac{N(s,a)}{N(s)}$.

## 2.1. A brief look at UCB values

Figure 1 illustrates the evolution of UCB values for a simple (one-player) bandit game with two slot arms, starting from a uniform prior policy. Each slot pays out a reward of $+1$ or $-1$; the probability of $+1$ is a fixed hidden parameter $p$. For one slot, $p = 0.6$, and for the other, $p = 0.4$. The inferior choice ($p = 0.4$) is never abandoned; this action is chosen $\Theta(1/\sqrt{N})$ of the time, where $N$ is the total number of simulations. As $N \to \infty$, the two UCB values get closer and closer together, approaching $0.2 + \Theta(1/\sqrt{N})$.

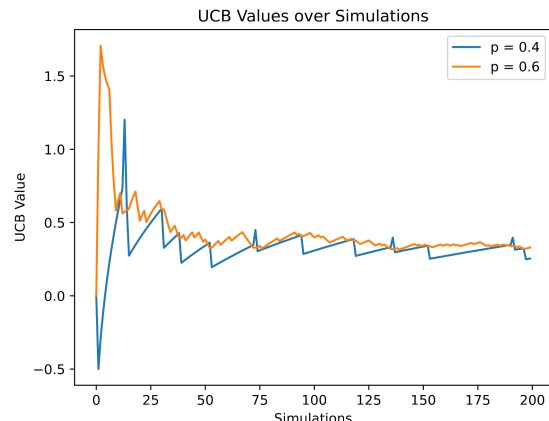

*Figure 1.* Illustration of UCB values over 200 simulations in bandit game with two slot arms.

In general, the UCB values for various actions tend to come closer and closer to each other as the number of simulations increases. The optimized posterior policy of (Grill et al., 2020), described next in section 3, is the unique policy in which the UCB values are all equal.

## 3. Regularized Policy Optimization

An alternative to $\hat{\pi}$ was explored in "Monte–Carlo tree search as regularized policy optimization" (Grill et al., 2020). They define the optimized posterior policy $\bar{\pi}(s, \cdot)$ to be the one maximizing the expected reward (assuming estimated rewards $Q(s, a)$) minus a penalty for diverging from the prior policy $\pi_0(s, \cdot)$. Specifically, $\bar{\pi}(s, \cdot)$ is the unique distribution on actions at game state $s$ which maximizes

$$\sum_a \bar{\pi}(s, a) Q(s, a) - \lambda \operatorname{KL-Div}(\pi_0(s, \cdot) \,||\, \bar{\pi}(s, \cdot))$$

where KL-Div is the Kullback-Leibler divergence, and $\lambda > 0$ is a constant. Taking $\lambda = \frac{C}{\sqrt{N(s)}}$, where $N(s)$ is the number of MCTS-UCB simulations at state $s$ and $C > 0$ is the exploration constant, they show that $\bar{\pi}$ agrees asymptotically to $\hat{\pi}$ (normalized visit counts) as $N(s)$ approaches infinity, but argue that $\bar{\pi}$ is superior to $\hat{\pi}$ when $N(s)$ is small. They suggest replacing $\hat{\pi}$ with $\bar{\pi}$ in the AlphaZero algorithm, where the $Q$-values are computed using the original MCTS-UCB variant of AlphaZero.

We note that the optimized posterior policy $\bar{\pi}$ has an interesting interpretation: it is essentially the policy which forces the UCB values of all actions to be equal. Recall that the UCB value is defined as

$$\operatorname{UCB}(s, a) = Q(s, a) + C \cdot \pi_0(s, a) \cdot \frac{\sqrt{N(s)}}{1 + N(s, a)}.$$

Now suppose that we have finished all the simulations at state $s$, and $\hat{\pi}(s, a) = \frac{N(s, a)}{N(s)}$ is the normalized visit counts. For simplicity, let us drop the "1+" in the denominator of the exploration term. Now we can rewrite the UCB value as

$$\operatorname{UCB}(s, a) = Q(s, a) + C \cdot \pi_0(s, a) \cdot \frac{\sqrt{N(s)}}{\hat{\pi}(s, a) N(s)}$$

$$= Q(s, a) + \frac{C}{\sqrt{N(s)}} \cdot \frac{\pi_0(s, a)}{\hat{\pi}(s, a)}.$$

Now replace $\hat{\pi}(s, a)$ with a variable $\bar{\pi}(s, a)$, and consider the problem of maximizing the objective function $F(\bar{\pi}(s, \cdot))$, defined as

$$\sum_a \bar{\pi}(s, a) Q(s, a) - \frac{C}{\sqrt{N(s)}} \operatorname{KL-Div}(\pi_0(s, \cdot) \,||\, \bar{\pi}(s, \cdot)),$$

subject to the constraint that $\sum_a \bar{\pi}(s, a) = 1$. A simple calculation shows that $\frac{\partial F}{\partial \bar{\pi}(s, a)} = \operatorname{UCB}(s, a)$. On the other hand, the constraint function $\sum_a \bar{\pi}(s, a) = 1$ has partial derivative 1 with respect to each variable $\bar{\pi}(s, a)$. Hence the method of Lagrange multipliers tells us that the optimum $\bar{\pi}(s, \cdot)$ can only occur where all the UCB values $\operatorname{UCB}(s, a)$ are equal. From Figure 1 one can see that the UCB values of

MCTS-UCB approach each other over time; this optimized posterior policy $\bar{\pi}$ forces them to be equal.

If $u$ is the common UCB value for the optimal posterior policy $\bar{\pi}(s, \cdot)$, then we have

$$\bar{\pi}(s, a) = \frac{C}{\sqrt{N(s)}} \frac{\pi_0(s, a)}{u - Q(s, a)}.$$

In particular, we are looking for the unique value of $u > \max_a Q(s, a)$ such that

$$\sum_a \frac{C}{\sqrt{N(s)}} \frac{\pi_0(s, a)}{u - Q(s, a)} = 1.$$

This value certainly exists, since as $u \to \max_a Q(s, a)^+$, the left-hand side approaches $+\infty$, whereas as $u \to +\infty$, the left-hand side approaches 0.

The optimal posterior policy $\bar{\pi}$ can be computed efficiently using Newton's method (cf. Algorithm 3). In this algorithm, the function $f$ is convex and since we begin with the initial value on the appropriate side of the solution, Newton's method is guaranteed to converge. See Appendix C for a proof of convergence.

## 4. Recursive MCTS (RMCTS)

An important feature of the optimized posterior policy $\bar{\pi}$ is that it can be computed locally at each game state $s$, using only the estimated action values $Q(s, \cdot)$ and the prior policy $\pi_0(s, \cdot)$. It does not require any details about the rest of the search tree. This is what permits us to define a recursive alternative to MCTS-UCB.

Recall that (Grill et al., 2020) suggests replacing the posterior policy $\hat{\pi}$ with the optimized posterior policy $\bar{\pi}$, where the estimated $Q$ values are gotten from the original MCTS-UCB variant of AlphaZero. Recursive MCTS (RMCTS) takes this suggestion much further by using optimized policies not only at the root state, but throughout the tree. The $Q$-values are computed in a recursive manner, where the value of nodes $s'$ in the search tree are approximated (recursively) by computing optimized posterior policies below $s'$. Hence RMCTS replaces the heuristic UCB formula with a rigorous, locally computable optimization problem that naturally aligns the search with the neural network's priors.

The search tree is generated by following prior network policies at each node (in particular it is not defined by UCB values). Each node $s$ in the tree consumes one simulation, and then awards the remaining simulations to its child actions according to the prior policy $\pi_0(s, \cdot)$. Thus if a state $s$ is afforded $N(s)$ simulations, then one simulation is used to acquire the prior policy $\pi_0(s, \cdot)$ and value $v_0(s)$ from the network, leaving $N(s) - 1$ simulations to be distributed among the available actions from $s$. The number of simulations $N(s, a)$ assigned to action $a$ (cf. Algorithm 2) has

expectation $\mathbb{E}[N(s,a)] = \pi_0(s,a)(N(s)-1)$, where $\pi_0$ is the prior network policy at state $s$. As in MCTS-UCB, if $s'$ is a nonterminal leaf (afforded one simulation), then its value is defined to be the network value $v_0(s')$. If $s'$ is a terminal state afforded $k$ simulations, then its value is the game score and all $k$ simulations are consumed there.

To make the counting precise: every non-terminal node in the RMCTS tree counts as exactly one simulation (one GPU inference call), and every terminal node awarded $k$ simulations consumes all $k$ of them. The total simulation count for any subtree therefore equals the number of non-terminal nodes it contains plus the total simulation credits assigned to terminal nodes within it. This definition is consistent with MCTS-UCB, where a simulation likewise ends upon visiting a non-terminal node for the first time (consuming one network inference), and revisiting a terminal node counts as a full simulation each time. See section 5 for a simple example illustrating RMCTS.

---

**Algorithm 1** RMCTS (for two-player zero-sum games)

1: **Input:** State $s$, number of simulations $N$, exploration constant $C$.
2: **Output:** Estimated value $\bar{v}$ and policy $\bar{\pi}$ at root state $s$
3: **if** $s$ is terminal **then**
4:     **return** score of $s$, NULL policy
5: **end if**
6: **Let** $\text{sgn}(s,t) = +1$ if same player is active for states $s$ and $t$, else $-1$
7: Acquire priors $v_0(s)$ and $\pi_0(s,\cdot)$ from neural network
8: $N(s,\cdot) \leftarrow \text{ASSIGN-SIMS}(s, N-1, \pi_0(s,\cdot))$
9: **Let** $A \leftarrow \{a : N(s,a) > 0\}$
10: **for** each action $a \notin A$ **do**
11:     $\bar{\pi}(s,a) \leftarrow 0$
12: **end for**
13: **for** each action $a \in A$ **do**
14:     **Let** $t$ be the state reached by taking action $a$ from $s$
15:     $v_t, \pi_t \leftarrow \text{RMCTS}(t, N(s,a), C)$
16:     $Q(s,a) \leftarrow \text{sgn}(s,t)\, v_t$
17: **end for**
18: **Let** $Q', \pi_0'$ be the restriction of $Q(s,\cdot)$ and $\pi_0(s,\cdot)$ to actions in $A$
19: $\pi' \leftarrow \text{OPTIMIZE-POLICY}(Q', \pi_0', N-1, C)$
20: **for** each action $a \in A$ **do**
21:     $\bar{\pi}(s,a) \leftarrow \pi'(a)$
22: **end for**
23: $\bar{v} \leftarrow \frac{1}{N} \cdot v_0(s) + \frac{N-1}{N} \cdot \sum_{a \in A} Q(s,a) \cdot \bar{\pi}(s,a)$
24: **return** $\bar{v}, \bar{\pi}$

---

**Algorithm 2** ASSIGN-SIMS

1: **Input:** State $s$, number of simulations $N$, prior policy $\pi_0(s,\cdot)$
2: **Output:** Number of simulations $N(s,a)$ assigned to each action $a$
3: Put all potential actions in some arbitrary order $a_1, a_2, \ldots, a_n$
4: $t_0 \leftarrow 0$
5: **for** $i = 1, 2, \ldots, n$ **do**
6:     $t_i \leftarrow N \sum_{j \leq i} \pi_0(s, a_j)$
7: **end for**
8: Generate a uniformly random number $x$ in $[0, 1)$
9: **for** $i = 1, 2, \ldots, n$ **do**
10:     $N(s, a_i) \leftarrow \#\{k \in \mathbb{Z} : t_{i-1} \leq x + k < t_i\}$
11: **end for**
12: **return** $N(s,a)$ for each action $a$

---

Algorithm 2 (ASSIGN-SIMS) randomly distributes a total of $N$ simulations to actions according to the prior policy $\pi_0(s,\cdot)$. Action $a$ is assigned $N(s,a)$ simulations. The expectation $\mathbb{E}[N(s,a)] = \pi_0(s,a)N$, and $\lfloor \pi_0(s,a)N \rfloor \leq N(s,a) \leq \lceil \pi_0(s,a)N \rceil$. We note that, in particular, every action $a$ is guaranteed to receive at least $\lfloor \pi_0(s,a)N \rfloor$ simulations. Finally, we have $\sum_a N(s,a) = N$.

---

**Algorithm 3** OPTIMIZE-POLICY

1: **Input:** estimated action rewards $Q$, prior policy $\pi_0$, number of simulations $N$, exploration constant $C$
2: **Output:** Updated policy $\bar{\pi}$ as defined in (Grill et al., 2020), maximizing

$$\sum_a \bar{\pi}(a)Q(a) - \frac{C}{\sqrt{N}} \text{KL-Div}(\pi_0 \,||\, \bar{\pi})$$

3: $\epsilon \leftarrow 10^{-10}$
4: $\lambda \leftarrow C/\sqrt{N}$
5: Define $f(u) \leftarrow -1 + \lambda \sum_a \frac{\pi_0(a)}{u - Q(a)}$
6: $u \leftarrow \max_a Q(a) + \lambda \pi_0(a)$
7: **while** $f(u) > \epsilon$ **do**
8:     $u \leftarrow u - \frac{f(u)}{f'(u)}$ {Newton's method converges here}
9: **end while**
10: **for** each action $a$ **do**
11:     $\bar{\pi}(a) \leftarrow \lambda \frac{\pi_0(a)}{u - Q(a)}$
12: **end for**
13: Normalize $\bar{\pi}$ so that $\sum_a \bar{\pi}(a) = 1$
14: **return** $\bar{\pi}$

---

The proof of convergence of Newton's method in Algorithm 3 appears in Appendix C.

The description of RMCTS given in Algorithm 1 is meant to be mathematically precise, and makes it clear why we

view RMCTS as a recursive algorithm. However, it is not efficient to implement this way. See Appendix B and https://github.com/bhoward73/rmcts for an efficient implementation of RMCTS. The efficient implementation generates the search tree iteratively, in a breadth-first manner. All nodes at the same depth form one large batch of GPU inferences, and so the GPU latency cost is largely mitigated. Contiguous memory is pre-allocated for all the nodes (and relevant data) for the search tree, improving cache performance.

The timings we report in section 6 are based on the efficient implementation of RMCTS described above. We compare the quality of RMCTS and MCTS-UCB in section 7, by pitting them against each other in three games: Connect-4, Dots-and-Boxes, and Othello. In this contest, both RMCTS and MCTS-UCB use the same neural network for priors.

## 5. A simple example

To illustrate how RMCTS works, we consider a simple one-player game. Although we have described RMCTS only in the context of two-player zero-sum games, it is clear how to adapt it to one-player games (single-agent decision processes). Indeed, the adaptation is straightforward: the active player is always the first player.

Consider the toy-sized one-player game illustrated in Figure 2, where the game tree is a binary tree with only two nonterminal states $s$ and $t$. The available actions are $\ell$ (left) and $r$ (right). Starting from the root state $s$, if we go left, we reach a terminal state with value 1. If instead we go right, we reach the nonterminal state $t$. From $t$, if we go left, we reach a terminal state with value $-3$. If we go right from $t$, we reach a terminal state with value 2.

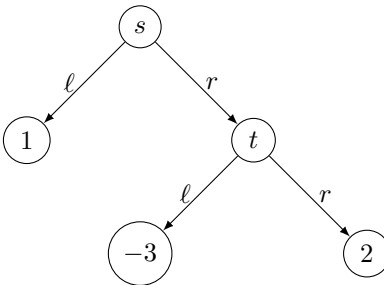

*Figure 2.* Clearly the best action from state $s$ in this one-player game is to go right, reaching state $t$, where we should again choose to go right, obtaining the maximal reward of 2.

Suppose that we are afforded $N = 1003$ simulations at the root state $s$. Initially we have no strong opinions and our prior policy is uniform, and our prior value is zero on all nonterminal states. Let's set our exploration constant to $C = 1$.

We spend one simulation on $s$ itself, for which our prior

value is $v_0(s) = 0$, leaving us with 1002 simulations to distribute among the two actions $\ell$ and $r$ from $s$. Since the prior policy is uniform we assign $N(s, \ell) = 501$ simulations to action $\ell$ and also $N(s, r) = 501$ simulations to action $r$. Since the left action $\ell$ from $s$ reaches a terminal state with value 1, we know the exact value $Q(s, \ell) = 1$ for the left action. For the right action $r$ from $s$, we first consume one simulation to obtain the uniform prior policy and zero prior value at state $t$. We assign $N(t, \ell) = 250$ simulations to action $\ell$ and $N(t, r) = 250$ simulations to action $r$ from state $t$.

Now that the search tree and simulation counts have been defined, we now turn to the recursive computation of $Q$ values. The optimized posterior policy at state $t$ (Algorithm 3) is approximately $\bar{\pi}(t, \ell) = 0.00445$ and $\bar{\pi}(t, r) = 0.996$. Giving one vote to the prior value $v_0(t) = 0$, our new estimated value $\bar{v}(t) = Q(s, r)$ is

$$\frac{1}{501} \cdot 0 + \frac{500}{501} \cdot (-3 \cdot 0.00445 + 2 \cdot 0.996) \approx 1.98.$$

Now that all estimated $Q$ values at state $s$ are known, we can compute the optimized posterior policy at $s$. The optimized posterior policy at the root state $s$ is computed (Algorithm 3) to be approximately $\bar{\pi}(s, \ell) = 0.016$ and $\bar{\pi}(s, r) = 0.984$. Thus, our final estimated value at the root state $s$ is

$$\frac{1}{1003} \cdot 0 + \frac{1002}{1003} \cdot (1 \cdot 0.016 + 1.98 \cdot 0.984) \approx 1.96.$$

Hence the posterior value of $s$ is approximately 1.96, and the posterior policy at $s$ strongly favors going right with a probability of 98.4%.

Note that it was very important to use recursion to define the $Q$ values. If we had assigned the $Q$ values to be the expected action values following the prior policy $\pi_0$ (no recursion), then our new posterior policy $\bar{\pi}$ would have favored going left from $s$, since going left yields a reward of 1, whereas the expected value of going right is $(-3 + 2)/2 = -0.5 < 1$.

## 6. Timings for MCTS-UCB vs RMCTS for three games

In this section we give timing comparisons of MCTS-UCB vs RMCTS for three games: Connect-4 (cf. (Allen, 2010)), Dots-and-Boxes (cf. (Berlekamp, 2000)), and Othello (cf. (Rose, 2005)). All timings use TensorRT (NVIDIA Corporation, 2024) to optimize the neural network inferences. We ran these tests on a desktop computer which has an NVIDIA RTX 3080 GPU, and an Intel i7-10700K CPU. All networks were implemented and trained using PyTorch (Paszke et al., 2019). In the cases of Dots-and-Boxes and Othello, the network was a ResNet with 8 residual blocks and 48 channels. For Connect-4, the network was a ResNet with 8 residual blocks and 64 channels. In all cases the kernel size was $3 \times 3$.

The main point of this paper is to address the latency bottleneck of neural network inferences in MCTS-UCB, especially in the intra-tree setting where we are searching a single root state. GPU inferences are the main bottleneck; game mechanics form an insignificant portion of the total computation time. The total time for a batch of inferences can be modeled as

$$\text{time} = \text{latency} + \frac{\text{batch size}}{\text{throughput}},$$

where latency is often a hundred to a thousand times larger than $1/\text{throughput}$. Thus the clearest route to achieving speedups is to increase the batch size, thereby amortizing the latency cost over more inferences. This is precisely what RMCTS accomplishes.

We first consider the case where we are timing MCTS for a single root state (the intra-tree setting). This situation applies when we (human) play against a pre-trained AI opponent, and we want to supplement the AI network with MCTS at each move. Here RMCTS has the greatest speed advantage over MCTS-UCB.

Table 1. Othello timings, one root state.

| $N$ | 64 | 256 | 1024 |
|---|---|---|---|
| MCTS-UCB | 25 ms | 98 ms | 390 ms |
| RMCTS | 2.5 ms | 4.2 ms | 8.9 ms |
| speedup | 10× | 24× | 44× |

Table 2. Dots-and-Boxes timings, one root state.

| $N$ | 64 | 256 | 1024 |
|---|---|---|---|
| MCTS-UCB | 25 ms | 93 ms | 370 ms |
| RMCTS | 4.7 ms | 6.7 ms | 8.0 ms |
| speedup | 5.2× | 15× | 48× |

Table 3. Connect-4 timings, one root state.

| $N$ | 64 | 256 | 1024 |
|---|---|---|---|
| MCTS-UCB | 35 ms | 150 ms | 550 ms |
| RMCTS | 4.9 ms | 9.2 ms | 12 ms |
| speedup | 7.2× | 16× | 45× |

We next consider the case when we are computing MCTS for a batch of root states (trees) at once: the inter-tree setting. This situation applies when we are using MCTS to generate rollouts for training the neural network. Here the latency cost of network inferences is largely mitigated for both MCTS-UCB and RMCTS, since the network inferences are done in batches in both cases. In MCTS-UCB, when

searching a batch of root states, we can move from one state to the next whenever a necessary inference prevents us from continuing the computation at the given state. Once we've passed through all states in the batch, then we wait for responses from the network before continuing. The latency cost is mitigated in this way, but RMCTS maintains a significant speed advantage, because the batch sizes for RMCTS are much larger, since an RMCTS batch consists of all nodes at a given depth across all search trees.

Table 4. Othello, 64 root states, average time per root state.

| $N$ | 64 | 256 | 1024 |
|---|---|---|---|
| MCTS-UCB | 1.2 ms | 5.3 ms | 27 ms |
| RMCTS | 0.33 ms | 1.1 ms | 4.2 ms |
| speedup | 3.5× | 4.7× | 6.6× |

Table 5. Dots-and-Boxes, 64 root states, average time per root state.

| $N$ | 64 | 256 | 1024 |
|---|---|---|---|
| MCTS-UCB | 1.1 ms | 4.6 ms | 23 ms |
| RMCTS | 0.22 ms | 0.57 ms | 1.7 ms |
| speedup | 4.9× | 8.2× | 13.7× |

Table 6. Connect-4, 64 root states, average time per root state.

| $N$ | 64 | 256 | 1024 |
|---|---|---|---|
| MCTS-UCB | 0.80 ms | 3.1 ms | 12 ms |
| RMCTS | 0.36 ms | 0.82 ms | 2.5 ms |
| speedup | 2.2× | 3.7× | 4.8× |

## 7. Comparing the quality of RMCTS vs MCTS-UCB

We find that RMCTS is at a small disadvantage if the number of simulations is equal. Most likely the reason for this is that MCTS-UCB creates the search tree in an adaptive manner, whereas RMCTS creates the search tree in a non-adaptive manner.

RMCTS trees are structurally shallower than MCTS-UCB trees for a fixed simulation budget: because the budget is distributed at each level proportionally to the prior across all children, the effective search depth scales as $O(\log_b N)$, where $b$ is the branching factor. MCTS-UCB, by contrast, can concentrate simulations along critical tactical lines and reach considerably greater depths. This is a deliberate trade-off — we exchange the high-variance deep tactical lines of MCTS-UCB for an exhaustive, low-variance evaluation that perfectly saturates GPU batching. Deeper but thinner

trees accelerate the value signal from terminal states at the cost of greater noise; spreading the budget according to the prior delays the value signal but reduces noise.

Using $\pi_0$ to guide allocation is further justified by the observation that, in practice, the policy signal converges faster than value estimates during training. This asymmetry is broadly recognized in reinforcement learning: recent methods such as Group Relative Policy Optimization (GRPO) (Shao et al., 2024) and Direct Preference Optimization (DPO) (Rafailov et al., 2023) sidestep value-network estimation entirely, relying on relative comparisons among alternatives rather than absolute value predictions. The underlying intuition is that judging which option is better is fundamentally easier than quantifying by exactly how much.

Although RMCTS is at a small disadvantage when the number of simulations is equal, it is at a significant advantage when the number of simulations for RMCTS is doubled. See Table 7 for an example of this in Othello. It seems a general rule of thumb that RMCTS wins games against MCTS-UCB when the number of simulations for RMCTS is twice that of MCTS-UCB, but RMCTS still takes far less time than MCTS-UCB even though it is using twice the number of simulations.

Table 7 shows the results of RMCTS vs MCTS-UCB in 64 games of Othello (32 played as first player, and 32 as second player). In both cases, MCTS-UCB had $N = 256$ simulations and required about 2.3 seconds per game. In the first batch of 64 games, RMCTS also had $N = 256$ simulations, but generally lost with a mean score (checker difference) of $-4.0$. The time required by RMCTS per game, however, was only 135 milliseconds. In the second set of 64 games, RMCTS was given $N = 512$ simulations, and this time it won with a mean score of 7.2. The mean time per game was 178 milliseconds in this case – still far less than the MCTS-UCB time of 2.3 seconds.

*Table 7.* Pitting RMCTS vs MCTS-UCB in 64 games of Othello. Doubling the number of simulations for RMCTS gives it a score advantage while maintaining a speed advantage.

| $N$ (for RMCTS)        | 256     | 512    |
|------------------------|---------|--------|
| mean score             | $-4.0$  | 7.2    |
| mean time per game     | 135 ms  | 178 ms |
| speedup over MCTS-UCB  | 17×     | 13×    |

To compare timings for training with 64 games at once, the average RMCTS time with $N = 512$ simulations is about 2.2 milliseconds, whereas the average time for MCTS-UCB with $N = 256$ simulations is about 5.3 milliseconds. Hence RMCTS has the advantage both in quality and time, with a speedup factor of 2.4×. Equating their performance, the speedup increases to above 3×. This is the kind of

comparison that matters most for training, and indeed this factor of 3× speedup in training time is what we observed in practice across the games we studied.

Recall from (Chaslot, 2010) and (Mirsoleimani et al., 2017) the concept of strength saturation using virtual loss: as the number of threads increases, eventually the playing strength levels out or even gets worse. This is not a problem for RMCTS; indeed, for environments having a finite number of states, the playing strength becomes perfect as the number of simulations goes to infinity (see Appendix D). Furthermore there is no duplication of work between threads – as the number of simulations increases (and thus the number of concurrent threads on the GPU), the search tree scales up naturally. Given a non-perfect neural network, we can see for example in Table 8 that strength continues to increase with the number of simulations.

*Table 8.* RMCTS vs RMCTS in Othello, where one player has $N$ simulations and the other has $N/2$.

| $N$ (for stronger player) | 128  | 256 | 512 | 1024 | 2048 |
|---------------------------|------|-----|-----|------|------|
| mean score                | 10.3 | 9.9 | 7.7 | 7.0  | 6.9  |

## 8. What next?

In the near future we plan to test an adaptive variant of RMCTS. This consists of splitting the simulation budget into $k$ re-explorations. After each re-exploration, we would store the values and policies at each node of the subtree explored so far. In the next iteration, the subtree goes deeper into branches favored by the posterior policies. The nodes that were previously explored would not count against the simulation budget since we would store their values and policies from the previous explorations. Thus the overall procedure becomes adaptive, and would likely outperform the non-adaptive version.

We would like to emphasize that RMCTS can be defined using various notions of optimized posterior policies. Other variants of optimized posterior policies can be considered, for example we can replace the Kullback-Leibler divergence with other divergence measures. Other divergences (eg. any Bregman (Bregman, 1967) divergence) could be used in place of the Kullback-Leibler divergence, and will guarantee policy improvement when $Q$ is the true value function (cf. appendix of (Grill et al., 2020)). For example, if we reverse the order of $\pi_0$ and $\bar{\pi}$ in the Kullback-Leibler divergence: i.e. $\bar{\pi}$ maximizes

$$\sum_a \bar{\pi}(s,a) Q(s,a) - \frac{C}{\sqrt{N(s)}} \text{KL-Div}(\bar{\pi} \parallel \pi_0),$$

then $\bar{\pi}(s,a) \propto \pi_0(s,a) \exp\left(\frac{\sqrt{N(s)}}{C} Q(s,a)\right)$. This is an

appealing formula, and easy to calculate; but it probably punishes the least-favored actions too much.

In general, RMCTS makes sense for reasonable variants of Algorithm 3, where the optimized posterior policy can be computed efficiently, and is computed based only on the estimated $Q$-values, prior policy, number of simulations, and exploration constant.

Recent advancements in regularized policy optimization for planning, most notably "Policy Improvement by Planning with Gumbel" (Danihelka et al., 2022), have introduced principled selection mechanisms like sequential halving to ensure consistent policy improvement. While such methods significantly enhance sample efficiency, they typically retain a sequential selection structure that requires awaiting neural network evaluations before allocating the next search budget. This results in significant hardware under-utilization when searching a single state, as the GPU remains idle between the small, sequential increments of the tree expansion. RM-CTS and Gumbel MuZero therefore address complementary efficiency axes: hardware efficiency (intra-tree parallelism) and sample efficiency (adaptive allocation via sequential halving), respectively. We have not directly compared the two methods in the experiments presented here; such a comparison, and a careful characterization of the regimes in which each dominates, is an important direction for future work. Finally, it would be interesting to try a hybrid approach that uses sequential halving at the root to minimize simple regret, but uses successive applications of RMCTS to evaluate the branches below the root.

We explored just three games: Connect-4, Dots-and-Boxes, and Othello. These are instructive benchmarks, but they are relatively small games with modest branching factors and short game lengths. Testing RMCTS at scale on Go, Chess, and Shogi — the canonical benchmarks for AlphaZero (Silver et al., 2018) — is an important direction, since the relative advantages of breadth-first versus adaptive search may shift substantially as the game tree grows.

Throughout this paper we have focused on applying RM-CTS to AlphaZero. However, RMCTS integrates seamlessly into the MuZero framework. Standard MuZero uses MCTS-UCB over a **learned latent space**: the true environment simulator is replaced by a learned dynamics model $g(s, a) = (r, s')$, and the prior policy and value are supplied by a prediction function $f(s) = (\pi_0, v_0)$. RMCTS can serve as a drop-in replacement for MCTS-UCB within MuZero, with its breadth-first expansion operating directly over these latent states and intermediate rewards without any other modification. Because RMCTS drastically increases the number of tree nodes evaluated per GPU-second through intra-tree batching, it acts as a much faster, fully-batched policy-improvement operator for the MuZero target network, potentially offering the same training-time speedups

observed in our AlphaZero-style experiments.

## 9. Conclusion

In this paper, we introduced RMCTS, a recursive approach to Monte–Carlo tree search that prioritizes hardware efficiency without sacrificing the theoretical rigor of regularized policy optimization. By adopting a non-adaptive, prior-guided tree expansion, RMCTS targets a different efficiency axis than methods like Gumbel MuZero with sequential halving (Danihelka et al., 2022): where Gumbel MuZero prioritizes **sample efficiency** through adaptive allocation, RMCTS prioritizes **hardware efficiency** through intra-tree batching. Both approaches rest on principled regularized-policy-optimization foundations; the question of which dominates in practice — and across which game domains — remains open and is an important direction for future work. We provided efficient code for RMCTS at `https://github.com/bhoward73/rmcts`, which also includes all the timing and quality comparisons presented in this paper.

Our approach addresses a limitation in MCTS-UCB-based implementations. While highly optimized libraries such as `mctx` (DeepMind et al., 2021) achieve impressive throughput via inter-tree batching across many environments, they remain latency-bound when searching a single state. RM-CTS provides a solution for this "single-root" regime by maximizing GPU utilization through intra-tree batching. As demonstrated in our experiments with Connect-4, Dots-and-Boxes, and Othello, this allows for speedups of over $40\times$ relative to MCTS-UCB, effectively enabling deep, high-quality search in real-time applications where only a single game instance is available. Extending these experiments to larger-scale games such as Go, Chess, and Shogi is an important next step.

Future work will explore adaptive variants of RMCTS that utilize multiple re-exploration iterations to refine the search tree while maintaining the benefits of batch inference. Directly benchmarking RMCTS against Gumbel MuZero (Danihelka et al., 2022) with sequential halving, and scaling experiments to games such as Go, Chess, and Shogi, are essential for establishing where each method's efficiency advantage dominates.

## Acknowledgements

We thank Tom Sznigir, James Barker, Timothy Chow, Emma Cohen, Ryan Eberhart, Stephen Fischer, Stephen Boyack, Ross Parker, and Lawren Smithline (all of IDA/CCR-Princeton) for helpful discussions and comments. We thank Laurent Sartran and Thomas Hubert for helpful discussions during the Google DeepMind presentation at the Institute for Advanced Study in Princeton, NJ in Spring 2025.

## Impact Statement

This paper presents work whose goal is to advance the field of Machine Learning, specifically in the domain of Monte Carlo Tree Search algorithms. The primary contribution is a computational efficiency improvement that makes game-playing AI systems faster and more accessible.

The potential societal impacts of this work are largely positive. By significantly reducing the computational cost of MCTS-based systems (achieving speedups of 40x or more in certain scenarios), this work could democratize access to strong AI opponents and training systems. Researchers and developers with limited computational resources will be better able to develop and deploy game-playing agents, which could accelerate research in planning, decision-making, and reinforcement learning more broadly.

The techniques developed here are domain-general and could be applied beyond two-player zero-sum games to other planning and decision-making problems where MCTS is applicable. This includes potential applications in robotics, autonomous systems, and strategic planning.

We do not anticipate specific negative societal consequences from this work. As with any advancement in AI capability, there is a general responsibility to ensure that powerful decision-making systems are developed with appropriate safeguards and ethical considerations, but this applies broadly to the field rather than specifically to the efficiency improvements presented here.

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

## A. MCTS-UCB

Here we give a precise description of the MCTS-UCB algorithm used in AlphaZero (Silver et al., 2018).

---

**Algorithm 4** MCTS-UCB

1: **Input:** State $s$, number of simulations $N$, exploration constant $C$
2: **Output:** New policy $\hat{\pi}$ at root state $s$
3: Given state $t$, let $\text{sgn}(t) = +1$ if player 1 to move in state $t$, else $-1$
4: visited $= \emptyset$
5: **while** $N > 0$ **do**
6: $\quad t \leftarrow s$
7: $\quad$ path $= []$
8: $\quad$ **while** $t$ is in visited and $t$ is nonterminal **do**
9: $\qquad$ Select action $a$ that maximizes $\text{UCB}(t, a) = Q(t, a) + C \cdot \pi_0(t, a) \frac{\sqrt{\sum_b N(t,b)}}{1+N(t,a)}$
10: $\qquad$ Append $(t, a)$ to path
11: $\qquad t \leftarrow$ state reached by taking action $a$ from $t$
12: $\quad$ **end while**
13: $\quad$ **if** $t$ is nonterminal **then**
14: $\qquad$ Add $t$ to visited
15: $\qquad Q(t, a) \leftarrow 0$ for all actions $a$
16: $\qquad N(t, a) \leftarrow 0$ for all actions $a$
17: $\qquad$ Acquire priors $v_0(t)$ and $\pi_0(t, \cdot)$ from neural network
18: $\qquad v \leftarrow \text{sgn}(t)v_0(t)$ (appropriate sign relative to player 1)
19: $\quad$ **else**
20: $\qquad v \leftarrow$ score of terminal state $t$, relative to player 1
21: $\quad$ **end if**
22: $\quad$ **for** $(t, a)$ in path **do**
23: $\qquad N(t, a) \leftarrow N(t, a) + 1$
24: $\qquad Q(t, a) \leftarrow Q(t, a) + \frac{\text{sgn}(t)v - Q(t,a)}{N(t,a)}$
25: $\quad$ **end for**
26: $\quad N \leftarrow N - 1$
27: **end while**
28: **for** action $a$ **do**
29: $\quad \hat{\pi}(a) \leftarrow \frac{N(s,a)}{\sum_b N(s,b)}$
30: **end for**
31: **return** $\hat{\pi}$

---

## B. RMCTS (Efficient Breadth-First Search Implementation)

The recursive description of RMCTS in Algorithm 1 is mathematically precise, but naïvely executed it calls the neural network once per tree node in sequence, leaving the GPU idle between calls. The breadth-first search (BFS) implementation described here processes all nodes at the same depth of the tree in a single neural-network batch, reducing the total number of batch calls from $O(N)$ to $O(\log N)$. We describe the general case of $m \geq 1$ simultaneously-searched root states ("lanes"); the single-root case is $m = 1$. A complete implementation is available at https://github.com/bhoward73/rmcts.

**Data representation.** All nodes are stored in a flat array indexed $0, 1, \ldots, M - 1$, where $M$ grows during Stage 1. The maximum possible value for $M$ is $mN$, assuming that each of the $m$ root states is afforded $N$ simulations. In practice we pre-allocate an array with enough capacity for $mN$ nodes. The $m$ root states occupy rows 0 through $m - 1$. Row $i$ records:

- game state $g_i$;

- prior policy $\pi_0^{(i)}(\cdot)$ and prior value $v_0^{(i)}$ (filled by the neural network, or set to the terminal reward for terminal states);

- per-action Q-values $Q_i(a)$ and visit counts $N_i(a)$, initialized to zero;

- parent index $p_i$ and action $a_i$ taken from the parent to reach node $i$;

- allocated simulation count $S_i$ and remaining count $R_i$, with $R_i$ initialized to $S_i$.

For a game state $g$, let $\mathrm{player}(g) \in \{+1, -1\}$ denote the active player. Note $\mathrm{sgn}(g_s, g_t) = \mathrm{player}(g_s) \cdot \mathrm{player}(g_t)$ in the notation of Algorithm 1. Two queues are maintained: $\mathcal{N}$ (nodes ready for expansion) and $\mathcal{I}$ (nodes awaiting neural-network inference). A node $i \geq m$ with $S_i = 1$ is called a **leaf**; a node with $S_i > 1$ and $g_i$ non-terminal is **internal**. Children are always allocated to rows with strictly higher index than their parent, so iterating rows in reverse order is guaranteed to visit every child before its parent.

---

**Algorithm 5** RMCTS-BFS

---

1: **Input:** non-terminal root states $g_0, \ldots, g_{m-1}$; simulations $N \geq 2$; constant $C$
2: **Output:** posterior policy $\bar{\pi}_i$ and value $\bar{v}_i$ for each root $i$
3: $M \leftarrow m$; $S_i \leftarrow N$, $R_i \leftarrow N$ for $i = 0, \ldots, m-1$
4: Acquire $\pi_0^{(i)}$ and $v_0^{(i)}$ from neural network for all roots simultaneously {initial batch}
5: $\mathcal{N} \leftarrow \{0, \ldots, m-1\}$; $\mathcal{I} \leftarrow \emptyset$
6: **while** $\mathcal{N} \neq \emptyset$ **do** {Stage 1: build the tree BFS level-by-level}
7:    FLUSH-NEW-STACK {expand $\mathcal{N}$; populate $\mathcal{I}$; Algorithm 6}
8:    **if** $\mathcal{I} = \emptyset$ **then**
9:       **break**
10:    **end if**
11:    Acquire $\pi_0^{(j)}$ and $v_0^{(j)}$ from neural network for all $j \in \mathcal{I}$ simultaneously {one batch per level}
12:    $\mathcal{N} \leftarrow \mathcal{I}$; $\mathcal{I} \leftarrow \emptyset$
13: **end while**
14: PROPAGATE-ALL {Stage 2: bottom-up value and policy computation; Algorithm 7}
15: **return** $\bar{\pi}_i$, $\bar{v}_i$ for $i = 0, \ldots, m-1$

---

**Algorithm 6** FLUSH-NEW-STACK

---

1: **Effect:** expands every node currently in $\mathcal{N}$; appends non-terminal children to $\mathcal{I}$
2: **while** $\mathcal{N} \neq \emptyset$ **do**
3:    Pop node $i$ from $\mathcal{N}$
4:    Restrict $\pi_0^{(i)}$ to valid actions in $g_i$ and renormalize {legalize prior}
5:    **if** $S_i = 1$ **then**
6:       **continue** {leaf: value is $v_0^{(i)}$; no children to create}
7:    **end if**
8:    $(k_a)_a \leftarrow$ ASSIGN-SIMS$(g_i, S_i - 1, \pi_0^{(i)})$ {Algorithm 2}
9:    **for** each action $a$ with $k_a > 0$ **do**
10:       $h \leftarrow \mathrm{nextState}(g_i, a)$
11:       $j \leftarrow M$; $M \leftarrow M + 1$ {allocate a new row}
12:       $g_j \leftarrow h$; $p_j \leftarrow i$; $a_j \leftarrow a$; $S_j \leftarrow k_a$; $R_j \leftarrow k_a$; $Q_j(\cdot) \leftarrow 0$; $N_j(\cdot) \leftarrow 0$
13:       **if** $h$ is terminal **then**
14:          $v_0^{(j)} \leftarrow$ terminal reward of $h$ for active player {no inference needed}
15:       **else**
16:          Push $j$ onto $\mathcal{I}$ {request neural-network inference}
17:       **end if**
18:    **end for**
19: **end while**

---

---

**Algorithm 7** PROPAGATE-ALL

---

1: **Effect:** propagates values bottom-up; writes root outputs $(\bar{\pi}_i, \bar{v}_i)$
2: **for** $i = M - 1, M - 2, \ldots, m$ **do** {children always precede their parents in this order}
3:     **if** $S_i = 1$ **then** {leaf (or terminal with $S_i = 1$): value is prior output}
4:         $v_i \leftarrow v_0^{(i)}$;   $\tau_i \leftarrow 1$
5:     **else if** $R_i = S_i$ **then** {terminal with $S_i > 1$: $R_i$ was never decremented by a child}
6:         $v_i \leftarrow v_0^{(i)}$;   $\tau_i \leftarrow S_i$
7:     **else** {internal: all children have called UPDATE-PARENT on $i$, so $R_i = 1$}
8:         $v_i \leftarrow$ COMPUTE-VALUE$(i)$;   $\tau_i \leftarrow S_i$ {Algorithm 8}
9:     **end if**
10:     UPDATE-PARENT$(p_i, a_i, v_i, \tau_i, \text{player}(g_i))$ {Algorithm 9}
11:     $R_i \leftarrow 0$
12: **end for**
13: **for** $i = 0, \ldots, m-1$ **do** {finalize each root}
14:     $(\bar{\pi}_i, \bar{v}_i) \leftarrow$ COMPUTE-VALUE$(i)$
15: **end for**

---

**Algorithm 8** COMPUTE-VALUE$(i)$

---

1: **Input:** internal node $i$ with $S_i \geq 2$
2: **Output:** posterior value $\bar{v}_i$; and posterior policy $\bar{\pi}_i$ when $i < m$
3: $A \leftarrow \{a : N_i(a) > 0\}$ {actions that received at least one simulation}
4: $Q_A \leftarrow (Q_i(a))_{a \in A}$
5: $\pi_{0,A}(a) \leftarrow \pi_0^{(i)}(a) \Big/ \sum_{b \in A} \pi_0^{(i)}(b)$ for each $a \in A$ {renormalize prior to $A$}
6: $\bar{\pi}_A \leftarrow$ OPTIMIZE-POLICY$(Q_A, \pi_{0,A}, S_i - 1, C)$ {Algorithm 3}
7: $v \leftarrow \sum_{a \in A} \bar{\pi}_A(a) \, Q_A(a)$
8: $\bar{v}_i \leftarrow v + \dfrac{v_0^{(i)} - v}{S_i}$ {one vote for the nnet prior value; cf. line 23 of Algorithm 1}
9: **if** $i < m$ **then** {root node: also record the full posterior policy}
10:     $\bar{\pi}_i(a) \leftarrow \bar{\pi}_A(a)$ for $a \in A$;   $\bar{\pi}_i(a) \leftarrow 0$ for $a \notin A$
11: **end if**
12: **return** $\bar{v}_i$ (and $\bar{\pi}_i$ if $i < m$)

---

**Algorithm 9** UPDATE-PARENT$(p, a, v_\text{c}, \tau_\text{c}, \text{pl}_\text{c})$

---

1: **Input:** parent row $p$; action $a$; child value $v_\text{c}$; child sim-count $\tau_\text{c}$; child active player $\text{pl}_\text{c} \in \{+1, -1\}$
2: $v \leftarrow \text{player}(g_p) \cdot \text{pl}_\text{c} \cdot v_\text{c}$ {$= \text{sgn}(g_p, g_\text{child}) \cdot v_\text{c}$: convert to parent's active-player frame}
3: $Q_p(a) \leftarrow \dfrac{Q_p(a) \cdot N_p(a) + v \cdot \tau_\text{c}}{N_p(a) + \tau_\text{c}}$ {update running average}
4: $N_p(a) \leftarrow N_p(a) + \tau_\text{c}$
5: $R_p \leftarrow R_p - \tau_\text{c}$

---

The three branches in PROPAGATE-ALL cover all node types. **Any** node with $S_i = 1$—whether a non-terminal leaf whose value came from the network, or a terminal state that received exactly one simulation—propagates with weight $\tau_i = 1$ and value $v_0^{(i)}$; the cases are handled identically. A **terminal** node with $S_i > 1$ is identified by $R_i = S_i$, since FLUSH-NEW-STACK never creates children for terminal states and therefore $R_i$ was never decremented; it propagates with full weight $\tau_i = S_i$. An **internal** node arrives in the reverse pass with $R_i = 1$: each child decremented $R_i$ by its $k_a$ via UPDATE-PARENT, for a total decrement of $S_i - 1$ (the budget distributed by ASSIGN-SIMS). COMPUTE-VALUE then

applies the same posterior-policy formula as the recursive algorithm (Algorithm 1, lines 16–23), but restricted to the actions in $A$ that actually received simulations.

## C. Convergence of Newton's method in Algorithm 3

Let

$$f(u) = -1 + \sum_a \frac{\lambda \pi_0(a)}{u - Q(a)},$$

where $u > \max_a Q(a)$. Let $Q_* = \max_a Q(a)$. As $u \to Q_*^+$, we have $f(u) \to +\infty$, and as $u \to +\infty$, we have $f(u) \to -1$. Since $f$ is continuous, by the intermediate value theorem there exists some $u_* > Q_*$ such that $f(u_*) = 0$. Note that $f'(u) = -\sum_a \frac{\lambda \pi_0(a)}{(u-Q(a))^2} < 0$. Therefore $f$ is strictly decreasing on the domain $u > Q_*$, and so $u_*$ is unique. The second derivative is $f''(u) = 2 \sum_a \frac{\lambda \pi_0(a)}{(u-Q(a))^3} > 0$. Thus $f$ is strictly convex on the domain $u > Q_*$.

Since $\bar{\pi}(a) = \frac{\lambda \pi_0(a)}{u_* - Q(a)} \le 1$ for all $a$, we have $u_* - Q(a) \ge \lambda \pi_0(a)$ for all $a$, and hence

$$u_* \ge \max_a Q(a) + \pi_0(a)\lambda.$$

We set our initial guess $u_0 = \max_a Q(a) + \lambda \pi_0(a)$, and so we have $Q_* < u_0 \le u_*$. With this initial guess and these properties of $f$, Newton's method converges monotonically to $u_*$ from below (cf. (Burden & Faires, 2011)). Moreover, because $f$ is smooth and $f'(u_*) \ne 0$, convergence is **quadratic**: if $e_n = u_* - u_n$ denotes the error at step $n$, then

$$\lim_{n\to\infty} \frac{e_{n+1}}{e_n^2} = -\frac{f''(u_*)}{2\,f'(u_*)} = \frac{\sum_a \dfrac{\lambda \pi_0(a)}{(u_* - Q(a))^3}}{\sum_a \dfrac{\lambda \pi_0(a)}{(u_* - Q(a))^2}} > 0,$$

where positivity follows from $f'' > 0$ and $f' < 0$. Because the initial guess already satisfies $0 < e_0 \le u_* - Q_*$ (a typically small interval), this strict quadratic convergence means that only a handful of iterations suffice to reach floating-point precision in practice. See Figure 3 for an illustration of this convergence.

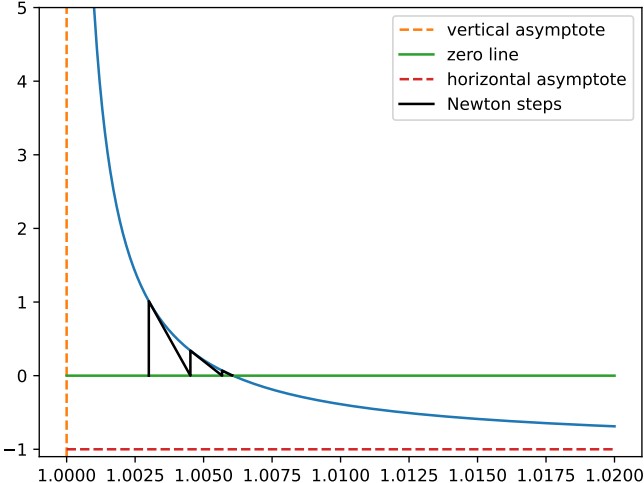

*Figure 3.* Illustration of Newton's method converging to the unique root $u_*$ of $f(u) = 0$. Here $Q_* = \max_a Q(a) = 1$. The function $f$ is strictly decreasing and strictly convex on the domain $u > Q_*$. Starting from any initial guess in $(Q_*, u_*]$, Newton's method converges monotonically to $u_*$ from below.

## D. Asymptotic Convergence of RMCTS to Perfect Play

We show that, in any finite two-player zero-sum game with a prior policy of full support, the policy and value returned by RMCTS converge to the minimax policy and value as the simulation budget approaches infinity. Let $\mathcal{T}$ denote the complete game tree rooted at initial state $s_0$, with maximum depth $D$. We assume $\mathcal{T}$ is a finite tree: every node is reached by exactly one sequence of actions from $s_0$, and every path terminates within $D$ steps. For any finite game, this can always be arranged by including the move count (number of actions taken since $s_0$) as part of the game state and terminating the game at depth $D$; distinct action sequences then correspond to distinct nodes even if the underlying game position repeats, and the depth of every node is well-defined. The argument proceeds in three steps: we first show that every node in $\mathcal{T}$ accumulates unboundedly many simulations as the root budget grows, then characterize the limit of OPTIMIZE-POLICY as the local count grows, and finally prove convergence to minimax play by downward induction on the remaining depth $D - d$.

**Lemma D.1** (Unbounded visits). *Suppose $\pi_0(s, a) > 0$ for every state $s \in \mathcal{T}$ and every legal action $a$ from $s$. As the root simulation budget $N \to \infty$, the simulation count $N(s) \to \infty$ for every node $s \in \mathcal{T}$.*

*Proof.* Fix any $s \in \mathcal{T}$ at depth $d$, reached via the unique path $P = (s_0 \xrightarrow{a_1} s_1 \xrightarrow{a_2} \cdots \xrightarrow{a_d} s_d = s)$ from the root. Define the **path probability**

$$p(P) = \prod_{i=1}^{d} \pi_0(s_{i-1}, a_i) > 0,$$

which is strictly positive by the full-support assumption. ASSIGN-SIMS (Algorithm 2) guarantees that if a node receives $M$ simulations, then any action $a$ receives either $\lfloor \pi_0(s, a)(M - 1) \rfloor$ or $\lceil \pi_0(s, a)(M - 1) \rceil$ simulations—always within 1 of $\pi_0(s, a)(M - 1)$. Hence the number of simulations $N(s_d)$ allocated to $s_d$ satisfies

$$
\begin{aligned}
N(s_d) &\geq (\cdots((N\pi_0(s_0, a_1) - 1)\pi_0(s_1, a_2) - 1)\cdots)\pi_0(s_{d-1}, a_d) - 1 \\
&= Np(P) - \sum_{j=2}^{d}\prod_{i=j}^{d} \pi_0(s_{i-1}, a_i) - 1 \\
&\geq Np(P) - d \\
&\geq Np(P) - D.
\end{aligned}
\tag{1}
$$

Since $p(P) > 0$ and $D$ is finite, $N(s) \to \infty$ as $N \to \infty$, regardless of the random choices made by ASSIGN-SIMS. $\qquad\square$

**Theorem D.2** (Minimax convergence). *Consider a two-player zero-sum game whose game tree $\mathcal{T}$ is a finite tree of maximum depth $D$ (this can always be arranged by including the move count in the game state, as described above). Suppose the prior policy satisfies $\pi_0(s, a) > 0$ for every state $s \in \mathcal{T}$ and every legal action $a$ from $s$. As the root simulation budget $N \to \infty$, the value $\bar{v}$ and policy $\bar{\pi}$ returned by RMCTS at the root converge to the minimax value and a minimax-optimal policy.*

The proof requires one further lemma characterizing the limit of OPTIMIZE-POLICY; we state and prove it now before returning to Theorem D.2.

**Notation.** Fix a node $s \in \mathcal{T}$ with $n$ legal actions, prior $\pi_0(s, a) > 0$, and local simulation count $N = N(s)$. Let $Q(a) = Q(s, a; N)$ denote the Q-values, which in general are random values whose distribution depends on $N$. The OPTIMIZE-POLICY subroutine (Algorithm 3) returns

$$\bar{\pi}(a) = \frac{\lambda\, \pi_0(a)}{u_* - Q(a)}, \qquad \lambda = \frac{C}{\sqrt{N}},$$

where $u_* > \max_a Q(a)$ is the unique root of $f(u) = -1 + \lambda \sum_a \frac{\pi_0(a)}{u - Q(a)} = 0$.

**Lemma D.3** (Posterior concentration). *Suppose the Q-values $Q(a; N)$ converge to limits $Q^*(a)$ as $N \to \infty$. Let $v^* = \max_a Q^*(a)$ and $A^* = \arg\max_a Q^*(a)$. Then as $N \to \infty$, the posterior policy $\bar{\pi}$ produced by OPTIMIZE-POLICY satisfies*

$$
\bar{\pi}(a) \to \begin{cases} \dfrac{\pi_0(a)}{\pi_0(A^*)} & a \in A^*, \\ 0 & a \notin A^*, \end{cases}
$$

*where $\pi_0(A^*) = \sum_{b \in A^*} \pi_0(b)$. In particular, $\bar{\pi}$ concentrates on the limiting Q-maximizing actions.*

*Proof.* For $a \notin A^*$, the limiting gap $v^* - Q^*(a) > 0$. Since $Q(a; N) \to Q^*(a)$, for all large enough $N$,

$$\max_b Q(b; N) - Q(a; N) \geq \tfrac{1}{2}\left(v^* - Q^*(a)\right) > 0.$$

Because $u_* > \max_b Q(b; N)$, this gives $u_* - Q(a; N) \geq (v^* - Q^*(a))/2 > 0$, so

$$\bar{\pi}(a) = \frac{\lambda \pi_0(a)}{u_* - Q(a; N)} \leq \frac{2\lambda \pi_0(a)}{v^* - Q^*(a)} \longrightarrow 0 \qquad (a \notin A^*),$$

since $\lambda = C/\sqrt{N} \to 0$. The normalization constraint $\sum_a \bar{\pi}(a) = 1$ then forces all remaining mass onto $A^*$. Within $A^*$, every action satisfies $Q^*(a) = v^*$, so $Q(a; N) - Q(a'; N) \to 0$ for all $a, a' \in A^*$. Hence $u_* - Q(a; N)$ is asymptotically equal for all $a \in A^*$, giving $\bar{\pi}(a)/\bar{\pi}(a') \to \pi_0(a)/\pi_0(a')$ within $A^*$. Since the total mass on $A^*$ tends to 1, the stated limit follows. $\qquad\square$

**Remark (Asymptotic visitation rate for all actions).** Treating Q-values as fixed at their current values, the analysis of Theorem D.3 yields a quantitative asymptotic that applies to **every** action simultaneously. Writing $u_* = Q^* + \delta$ and using $f(u_*) = 0$ with $\lambda = C/\sqrt{N}$, to leading order the terms with $a \notin A^*$ contribute $O(\lambda)$ to the normalization sum, so

$$\frac{\lambda \pi_0(A^*)}{\delta} = 1 - O(\lambda) \implies \delta \approx \frac{C \pi_0(A^*)}{\sqrt{N}}.$$

Substituting $u_* - Q(a) = \delta + Q^* - Q(a)$ into $\bar{\pi}(a) = \lambda \pi_0(a)/(u_* - Q(a))$ and simplifying gives the uniform asymptotic

$$\bar{\pi}(a) \approx \frac{\pi_0(a)}{\dfrac{\sqrt{N}}{C}\left(Q^* - Q(a)\right) + \pi_0(A^*)} \qquad (N \to \infty),$$

where $Q^* = Q(A^*)$ and $\pi_0(A^*) = \sum_{b \in A^*} \pi_0(b)$. This single formula captures both regimes: setting $Q^* - Q(a) = 0$ (i.e. $a \in A^*$) recovers the limit $\pi_0(a)/\pi_0(A^*)$ from Theorem D.3, while for $a \notin A^*$ the dominant term in the denominator is $(\sqrt{N}/C)(Q^* - Q(a))$, so

$$\bar{\pi}(a) \approx \frac{C \pi_0(a)}{\sqrt{N}\left(Q^* - Q(a)\right)} = O\left(\tfrac{1}{\sqrt{N}}\right).$$

Thus inferior actions are never completely abandoned; they continue to be visited at a rate of $\Theta(1/\sqrt{N})$, analogous to UCB-style exploration.

*Proof of Theorem D.2.* By Lemma D.1, $N(s) \to \infty$ for every $s \in \mathcal{T}$ as $N \to \infty$. We prove by downward induction on the remaining depth $D - d$ that for every non-terminal $s \in \mathcal{T}$ at depth $d$, as $N \to \infty$:

$$\bar{v}(s) \to v^*(s) \qquad \text{and} \qquad \bar{\pi}(s, \cdot) \to \bar{\pi}^*(s, \cdot),$$

where $v^*(s)$ is the minimax value and $\bar{\pi}^*$ is any minimax-optimal policy.

*Base case ($D - d = 1$, i.e. $d = D - 1$).* Every child of $s$ lies at depth $D$ and is therefore terminal, so $Q(s, a)$ is the exact game score with no approximation. Since $N(s) \to \infty$, Lemma D.3 (with fixed $Q^*(a) = Q(s, a)$) gives $\bar{\pi}(s, \cdot) \to \bar{\pi}^*(s, \cdot)$, and

$$\bar{v}(s) = \frac{v_0(s)}{N(s)} + \frac{N(s) - 1}{N(s)} \sum_a \bar{\pi}(s, a) Q(s, a) \longrightarrow v^*(s),$$

since the first term vanishes and the second converges to the minimax value.

*Inductive step ($D - d > 1$).* Assume the claim holds for every non-terminal state at depth $> d$. Let $s$ be a non-terminal state at depth $d$.

**(i) Q-values converge.** Each action $a$ leads to child $s_a$, which receives $N(s_a) = \pi_0(s, a)(N(s) - 1) + O(1) \to \infty$. If $s_a$ is terminal, $Q(s, a) = v^*(s, a)$ exactly. Otherwise, the inductive hypothesis gives $\bar{v}(s_a) \to v^*(s_a)$, hence

$$Q(s, a) = \text{player}(s) \cdot \text{player}(s_a) \cdot \bar{v}(s_a) \longrightarrow v^*(s, a).$$

**(ii) Posterior policy concentrates.** Since $Q(s, a) \to v^*(s, a)$ for every $a$ and $N(s) \to \infty$, Lemma D.3 gives $\bar{\pi}(s, \cdot) \to \bar{\pi}^*(s, \cdot)$.

**(iii) Node value converges.**

$$\bar{v}(s) \;=\; \frac{v_0(s)}{N(s)} \;+\; \frac{N(s) - 1}{N(s)} \sum_a \bar{\pi}(s, a) \, Q(s, a) \;\longrightarrow\; v^*(s),$$

since the first term vanishes and the second converges because both $\bar{\pi}(s, \cdot)$ and $Q(s, \cdot)$ converge to their minimax counterparts.

This completes the induction. □

**Remark.** The full-support assumption on $\pi_0$ is used in Lemma D.1 to ensure $p(P) > 0$ for every path $P$ in $\mathcal{T}$, which by (1) guarantees $N(s) \to \infty$ for every $s \in \mathcal{T}$. Since we always begin training with policies of full support (in fact, often uniform), and the policy optimization includes the regularization term $\lambda \sum_a \pi_0(a) \log \pi(a)$, the support of the policy is never reduced during training. So the full-support assumption is satisfied by any trained network.

**Remark.** We point out that although RMCTS converges to perfect play as $N \to \infty$, it certainly not guaranteed to converge to perfect play in the AlphaZero training loop, where a fixed number of simulations is used at each training step, but the number of training steps grows without bound. This is a flaw of the AlphaZero training loop itself, not of RMCTS in particular; the same issue applies to MCTS-UCB. However, we observe that RMCTS-trained networks do converge to strong play (typically super-human) in our experiments.

## E. Round-Robin Training Comparisons

We present round-robin tournament results comparing networks trained with MCTS-UCB versus RMCTS. Since the main point of this paper is to introduce RMCTS and demonstrate its practical speedup over MCTS-UCB, we do not attempt to optimize the training loop or hyperparameters for either method; instead, we simply use the same (reasonable) settings for both methods (except that RMCTS uses twice as many simulations, because this is where the quality is about the same as MCTS-UCB) and show that RMCTS reaches a given level of ability in about $1/3$ the training time as MCTS-UCB, as expected from the fact that it sees about $3\times$ more training data in the same amount of wall-clock time with these settings.

For each game, networks **m1**–**m6** were trained using MCTS-UCB with $N = 64$ simulations and networks **r1**–**r6** were trained using RMCTS with $N = 128$ simulations; both sets used a batch of 64 simultaneous root states to amortize GPU latency. With these settings, the RMCTS networks see about $3\times$ more training data in the same amount of wall-clock time than do the MCTS-UCB networks, whereas the quality of the tree search is similar. So we expect them to reach a given level of ability in about $1/3$ the training time, and the tables here indicate that is indeed the case.

The index denotes the number of hours of training (e.g., **m5** is the MCTS-UCB network after 5 hours). At evaluation time each network policy is strengthened by MCTS-UCB tree search (even the RMCTS-trained networks) using $N = 64$ simulations. The action is selected randomly but uses a cold temperature $\tau = 0.2$; i.e. the posterior policy is raised to the power $1/\tau = 5$ and renormalized before sampling. Bold entries highlight the key cross-method comparisons that support the $3\times$ training-time speedup of RMCTS over MCTS-UCB observed in our experiments.

*Table 9.* Round-robin scores for Othello. Here scores are averaged over 200 games per pair, with each network playing first for 100 games and second for 100 games. **Bold**: m3 loses to r1 despite 3× more training; m6 loses to r2 as well. We suspect the lack of transitivity in the r1-r6 networks has something to do with the complexity of Othello; sometimes the networks can get off-policy from their earlier training data and end up in a different local optimum, which can lead to non-transitive behavior.

|     | m1    | m2     | m3     | m4     | m5     | m6     | r1       | r2       | r3      | r4      | r5      | r6      |
| --- | ----- | ------ | ------ | ------ | ------ | ------ | -------- | -------- | ------- | ------- | ------- | ------- |
| m1  |       | −42.20 | −44.62 | −42.46 | −42.30 | −40.76 | −41.98   | −42.49   | −39.51  | −39.73  | −38.92  | −35.36  |
| m2  | 42.20 |        | −20.32 | −23.86 | −26.87 | −25.19 | −26.57   | −28.51   | −25.80  | −26.34  | −22.18  | −22.82  |
| m3  | 44.62 | 20.32  |        | −9.40  | −13.77 | −14.51 | **−14.04** | −15.71 | −16.91  | −14.02  | −11.01  | −8.96   |
| m4  | 42.46 | 23.86  | 9.40   |        | −7.55  | −9.80  | −4.79    | −15.87   | −13.94  | −13.77  | −9.43   | −7.60   |
| m5  | 42.30 | 26.87  | 13.77  | 7.55   |        | −8.21  | −0.57    | −20.16   | −20.98  | −9.23   | −6.21   | −8.96   |
| m6  | 40.76 | 25.19  | 14.51  | 9.80   | 8.21   |        | 2.52     | **−19.65** | −13.41 | −10.59  | −4.35   | −3.79   |
| r1  | 41.98 | 26.57  | **14.04** | 4.79 | 0.57   | −2.52  |          | −10.80   | −5.42   | −4.29   | 1.42    | 1.84    |
| r2  | 42.49 | 28.51  | 15.71  | 15.87  | 20.16  | **19.65** | 10.80 |          | −1.97   | −3.09   | −4.41   | −1.21   |
| r3  | 39.51 | 25.80  | 16.91  | 13.94  | 20.98  | 13.41  | 5.42     | 1.97     |         | −4.98   | −9.56   | −2.95   |
| r4  | 39.73 | 26.34  | 14.02  | 13.77  | 9.23   | 10.59  | 4.29     | 3.09     | 4.98    |         | −9.90   | 0.61    |
| r5  | 38.92 | 22.18  | 11.01  | 9.43   | 6.21   | 4.35   | −1.42    | 4.41     | 9.56    | 9.90    |         | −8.75   |
| r6  | 35.36 | 22.82  | 8.96   | 7.60   | 8.96   | 3.79   | −1.84    | 1.21     | 2.95    | −0.61   | 8.75    |         |

*Table 10.* Round-robin scores for Dots-and-Boxes. Here scores are averaged over 200 games per pair, with each network playing first for 100 games and second for 100 games. **Bold**: Here r1 (column) does lose to m3 (row) but it is a close call; r2 does beat m6.

|     | m1    | m2    | m3     | m4     | m5    | m6    | r1      | r2       | r3     | r4     | r5     | r6     |
| --- | ----- | ----- | ------ | ------ | ----- | ----- | ------- | -------- | ------ | ------ | ------ | ------ |
| m1  |       | −8.89 | −10.37 | −10.74 | −10.36 | −10.74 | −8.55 | −10.62 | −10.24 | −10.71 | −10.66 | −10.35 |
| m2  | 8.89  |       | −5.89  | −6.90  | −7.59 | −7.41 | −2.96   | −7.59    | −8.33  | −8.63  | −8.73  | −8.37  |
| m3  | 10.37 | 5.89  |        | −3.43  | −4.82 | −5.19 | **0.46** | −6.34 | −7.27  | −7.28  | −7.66  | −7.23  |
| m4  | 10.74 | 6.90  | 3.43   |        | −2.78 | −3.19 | 2.01    | −3.82    | −5.67  | −5.92  | −6.14  | −5.93  |
| m5  | 10.36 | 7.59  | 4.82   | 2.78   |       | −1.32 | 3.78    | −2.79    | −5.20  | −5.89  | −5.69  | −5.69  |
| m6  | 10.74 | 7.41  | 5.19   | 3.19   | 1.32  |       | 3.80    | **−2.68** | −4.96 | −5.07  | −5.66  | −5.13  |
| r1  | 8.55  | 2.96  | **−0.46** | −2.01 | −3.78 | −3.80 |        | −7.92    | −8.41  | −8.16  | −8.23  | −8.05  |
| r2  | 10.62 | 7.59  | 6.34   | 3.82   | 2.79  | **2.68** | 7.92 |          | −3.26  | −4.21  | −4.79  | −4.65  |
| r3  | 10.24 | 8.33  | 7.27   | 5.67   | 5.20  | 4.96  | 8.41    | 3.26     |        | −1.57  | −1.96  | −2.47  |
| r4  | 10.71 | 8.63  | 7.28   | 5.92   | 5.89  | 5.07  | 8.16    | 4.21     | 1.57   |        | −1.30  | −1.74  |
| r5  | 10.66 | 8.73  | 7.66   | 6.14   | 5.69  | 5.66  | 8.23    | 4.79     | 1.96   | 1.30   |        | −0.62  |
| r6  | 10.35 | 8.37  | 7.23   | 5.93   | 5.69  | 5.13  | 8.05    | 4.65     | 2.47   | 1.74   | 0.62   |        |

*Table 11.* Round-robin scores for Connect Four. Here scores are accumulated over 100 games per pair, with each network playing first for 50 games and second for 50 games. **Bold**: m3 loses to r1 despite 3× more training; and m6 loses to r2.

|     | m1  | m2  | m3    | m4  | m5  | m6    | r1      | r2    | r3   | r4   | r5   | r6   |
| --- | --- | --- | ----- | --- | --- | ----- | ------- | ----- | ---- | ---- | ---- | ---- |
| m1  |     | −39 | −67   | −74 | −77 | −74   | −62     | −70   | −83  | −93  | −90  | −97  |
| m2  | 39  |     | −38   | −31 | −51 | −67   | −7      | −69   | −79  | −82  | −94  | −88  |
| m3  | 67  | 38  |       | −20 | −6  | −40   | **−8**  | −51   | −75  | −90  | −87  | −91  |
| m4  | 74  | 31  | 20    |     | −9  | −17   | 2       | −38   | −59  | −83  | −82  | −83  |
| m5  | 77  | 51  | 6     | 9   |     | −6    | −7      | −38   | −58  | −79  | −82  | −87  |
| m6  | 74  | 67  | 40    | 17  | 6   |       | 16      | **−44** | −62 | −77  | −76  | −83  |
| r1  | 62  | 7   | **8** | −2  | 7   | −16   |         | −69   | −77  | −96  | −88  | −96  |
| r2  | 70  | 69  | 51    | 38  | 38  | **44** | 69     |       | −51  | −66  | −71  | −81  |
| r3  | 83  | 79  | 75    | 59  | 58  | 62    | 77      | 51    |      | −38  | −50  | −61  |
| r4  | 93  | 82  | 90    | 83  | 79  | 77    | 96      | 66    | 38   |      | −8   | −22  |
| r5  | 90  | 94  | 87    | 82  | 82  | 76    | 88      | 71    | 50   | 8    |      | −5   |
| r6  | 97  | 88  | 91    | 83  | 87  | 83    | 96      | 81    | 61   | 22   | 5    |      |

