# OpenReview forum: "Recursive Monte-Carlo Tree Search"
_ICML.cc/2026/Conference — ICML 2026 regular_

### Official Review · Reviewer_EUZM · 2026-02-20

**Soundness:** 2
**Presentation:** 4
**Significance:** 2
**Originality:** 3
**Overall Recommendation:** 4
**Confidence:** 4

**Summary:**

The authors propose a recursive variant of Monte-Carlo Tree Search that serves as a drop-in replacement for MCTS used in AlphaZero. The motivation is, original MCTS is sequential, lacking parallelism, which leads to higher wall-clock times. Recursive MCTS circumvents this by building the search tree in a non-adaptive fashion, making the nodes at the same depth independent of each other, and thus allowing for parallelism which can be exploited by modern GPUs, improving wall-clock time.

**Compliance With Llm Reviewing Policy:**

Affirmed.

**Final Justification:**

The authors have adequately answered my questions in the rebuttal process, and hence I raised my score to weak accept.

**Key Questions For Authors:**

Please look at Weaknesses. I would increase my score if there is more justification to the depth of the tree, adaptive nature of the tree, and qualitative results (Reward scores of other environments).

**Limitations:**

Yes.

**Strengths And Weaknesses:**

Strengths:
1. Massive wall-clock speed-ups for RMCTS over the traditional MCTS. Takes a very simple idea of regularised policy optimisation and scales it to RMCTS taking advantage of the fact all policies and Q-values can be computed locally due to static nature of the tree.
2. The algorithm is very simple to implement. It's just a one-line drop into the sequential MCTS. The anonymous repository is quite well implemented and very easy to follow.
3. The paper is well written. Unlike most papers written today, the abstract is kept short (3 lines), the introduction and the technical chapters directly get to the point, making the paper very easy to read.

Weaknesses:
1. There is no discussion on tree depth. Because the simulation budget is split among all children at each level proportionally to the prior, the effective depth scales as roughly log_b(N) where b is the branching factor. MCTS-UCB can reach depth 20-30 along critical tactical lines, while RMCTS with the same budget produces a much shallower tree. Higher tree depths allow for better value estimations, equivalent to "deep planning". The paper never reports or analyzes tree depths, which is a significant omission.
2. Experimental evaluation is narrow in scope. Only three small games (Connect-4, Dots-and-Boxes, Othello) are tested, all with tiny networks (8 residual blocks, 48-64 channels). There is no evidence that the approach scales to complex domains like Go or chess with larger networks and higher branching factors. This also ties back to there being no discussion on depth of the tree and actions.
3. Non-adaptive tree expansion can waste budget on unpromising branches. RMCTS commits its tree structure entirely based on the prior policy, with no mechanism to redirect simulations toward promising actions during search. When the prior is inaccurate particularly during early training or new states, a significant fraction of the budget can be lost. The paper only evaluates with already-trained networks and makes no assumptions on the quality of the priors.
4. Training claims (Quality of RMCTS over MCTS) are unsupported . The authors claim RMCTS-trained networks match MCTS-UCB-trained networks in one-third the training time, but provide no training curves, to substantiate this. There is only comparison of mean-scores between Sequential-MCTS and R-MCTS on Othello, and not on other environments.
5. The "2x simulations" rule of thumb lacks formal justification. This ratio is observed empirically in one game and presented as a general guideline, but it almost certainly depends on the prior quality, branching factor, and simulation budget. No analysis is provided for when this ratio might be significantly worse.
6. The most promising extension is unimplemented. The adaptive re-exploration variant described in Section 8, which could address the depth and non-adaptivity limitations, is left entirely to future work. This is arguably the version that would make RMCTS genuinely competitive in complex domains in my opinion.

---

> ### Author Rebuttal · Authors · 2026-03-31
>
> We thank the reviewer for their feedback.
> We address the points they raise below.
>
> Tree Depth:
> While RMCTS trees are structurally shallower for a fixed simulation
> count due to $\log_b(N)$ breadth-first expansion, this is a deliberate trade-off.
> We exchange the high-variance, deep tactical lines of MCTS-UCB for an exhaustive,
> low-variance local evaluation that perfectly saturates GPU batching.
> Given equal wall-clock time, the massive simulation advantage of RMCTS
> mitigates the depth discrepancy.
>
> We can add an analysis of depth metrics to the appendix, or
> at the very least acknowledge the shallow depth issue in the main body.
>
>
> Scope and Scaling:
> We clarify that the $3\times$ training speedup and $2\times$ simulation
> equivalence are empirical findings strictly bounded by the environments tested.
> While evaluating on Go or Chess is outside the scope of this initial work,
> the theoretical justification for RMCTS scaling to such complex domains relies
> on hardware architecture. As model sizes increase and must be distributed across
> multiple GPUs (e.g., multi-H100 nodes), communication latency heavily bottlenecks
> sequential MCTS-UCB. RMCTS is explicitly designed to bypass this latency floor by
> fully saturating parallel throughput, suggesting these efficiency gains will
> persist or even compound in larger regimes. We will clarify these empirical
> boundaries and theoretical scaling properties in the camera-ready version.
>
>
> Nonadaptive Tree Expansion:
> We agree that the nonadaptive expansion is an issue. However,
> if we assume that the network has been trained by RMCTS itself,
> then there is a very strong correlation between the prior policy
> and the values. If the prior policy is very bad, then it seems
> reasonable that the values are likewise untrustworthy.
> We have found in practice, that during
> training, the value loss is considerably higher than the policy loss.
> This is true both for RMCTS as well as MCTS-UCB. So we felt that
> trusting the policy was a reasonable way to spread attention.
> Ultimately, we must depend on the true values at terminal game states
> to percolate their way up the tree over time. If we have deep
> but thin trees, we accelerate the reward signal but also suffer from
> greater noise. If we spread out the budget by
> following the prior policy, then we delay the reward signal (shallower tree)
> but also have less noise. One way to address a bad prior would
> of course be to implement the multi-stage version of RMCTS (which
> we have not yet finished.)
>
>
> Training Claims:
> We attached a table for Connect-4 to our response to ecQ2; this table
> shows the results of pitting 12 different networks against each other.
> The first six were trained with MCTS-UCB and are snapshots
> after 1, 2, 3, 4, 5, and 6 hours of training.
> The last six were trained with RMCTS (same hyperparameters)
> and are again snapshots at 1, 2, 3, 4, 5, and 6 hours.
> Each contest begins at the canonical starting position,
> uses a cold temperature to select actions (so somewhat close to
> taking argmax of the posterior policy). There were 100 games played in
> each case, with 50 games as player 1 and 50 games as player 2.
> In all cases, the exploration used MCTS-UCB with 64 simulations.
>
> We note that the 3rd hour MCTS-UCB network loses (score -8) to
> the 1st hour RMCTS network. Similarly, the 6th hour MCTS-UCB network loses
> (score -44) to the 2nd hour RMCTS network. This agrees with our
> claim of 3X speedup in training.
>
>
> Implementing Multi-Stage RMCTS:
> We do agree that for competition mode (i.e. winning games) that
> this is probably superior. There are nuances of
> how to do it correctly, which we are still working on and we're not
> ready to include in the present paper. However, for training,
> it is less intuitive that it would be better than vanilla one-stage RMCTS.
> Introducing this multi-stage variant does introduce complications.
> We feel that exposing the simplest form of the algorithm in a timely fashion has a lot of merit.
> After all, in the grand scheme of things, RMCTS is only an idea,
> and no doubt will be improved in myriad ways by various researchers.
> For that reason, we felt the simplest form of the algorithm is the best
> one to communicate.

---

> > ### Author Rebuttal · Reviewer_EUZM · 2026-03-31
> >
> > My questions have been adequately addressed by the authors, and would like to raise my score.

---

### Official Review · Reviewer_1Atp · 2026-02-24

**Soundness:** 3
**Presentation:** 3
**Significance:** 4
**Originality:** 3
**Overall Recommendation:** 4
**Confidence:** 3

**Summary:**

This paper introduces Recursive Monte‑Carlo Tree Search (RMCTS), a drop‑in replacement for AlphaZero‑style MCTS-UCB that replaces sequential tree traversal with breadth‑first, fully parallelizable expansion. The key idea is to recursively compute optimized posterior policies (from Grill et al., 2020) at every node in the tree, rather than only at the root. RMCTS uses a non-adaptive prior-driven exploration strategy to define the entire search tree structure, after which it performs a bottom‑up dynamic‑programming‑like evaluation where the posterior policy at each node is computed by solving a convex optimization problem that equalizes UCB values. Because all nodes at a given depth can be batched, RMCTS significantly increases GPU utilization and mitigates latency bottlenecks typical in MCTS-UCB’s sequential selection-inference cycle. The authors provide an efficient implementation and benchmarks on Connect‑4, Dots-and-Boxes, and Othello, showing large speedups (up to 40×) with competitive playing strength given modestly increased simulation budgets.

**Compliance With Llm Reviewing Policy:**

Affirmed.

**Final Justification:**

I have raised my scores for soundness and presentation as well as the overall score following the authors' rebuttal. RMCTS is an elegant algorithm that successfully addresses GPU latency bottlenecks in MCTS-UCB via breadth-first, fully parallelizable expansion.

**How the Rebuttal Addressed My Concerns:**

*Definitions*: The authors provided a clear, MCTS-aligned definition of "simulations" (counting non-terminal nodes and terminal visit counts) to be included in the final text.

*Theoretical Soundness*: They effectively proved that the divergence penalty's strict convexity guarantees convergence and concentration on Q-maximal actions despite the initial $\pi_0$ bias. The proposed anchored "multi-stage" variant practically resolves stability issues.

*Training Integration*: The rebuttal clearly outlined how the RMCTS posterior policy and weighted values feed into the self-play loop.

*Experimental Fairness*: The authors clarified that the 64-root baseline already achieved full GPU saturation, validating the wall-clock speedup comparisons against standard MCTS-UCB.

**Conclusion:**

The authors constructively closed all theoretical and descriptive gaps. Provided the rebuttal clarifications are incorporated into the camera-ready manuscript, this is a technically sound, practically impactful paper that is a strong addition to the conference.

**Key Questions For Authors:**

1. Definition and interpretation of “simulations”
Can you clarify why RMCTS calls each NN inference a “simulation,” and how this corresponds to the semantics of MCTS-UCB simulations?


2. Theoretical mismatch between expected visit counts and true UCB visitation
Since RMCTS uses $\pi_0(s,·)$ rather than the posterior policy to allocate $N(s,a)$, the expected visit counts do not correspond to MCTS-UCB. Does this bias matter? Can you provide an ablation? What happens if I run multiple iterations of RMCTS and replace the prior policy with the current policy in each? Does this fixed-point iteration converge? Does it remove the bias and improve sample efficiency in training?


3. Integration with training loops
How should RMCTS be used during self‑play training? Are leaf nodes actually evaluated with the value network each iteration? Or should RMCTS be rather considered as a test-time scaling method? If so, a clearer discussion of intended operation would benefit the paper.


4. Role of exploration when the network is fixed
In the single-root inference scenario, what is the rationale for maintaining exploration terms? Why not use a standard policy iteration technique instead? Is there a theoretical rationale?


5. Fair comparison to MCTS-UCB in training settings
Why not increase the number of parallel root states for MCTS-UCB? Why does MCTS-UCB need the same number of root states as RMCTS? Does this change speedup factors?


Clarifying these would significantly improve the paper’s credibility and practical relevance. In particular, I am willing to raise my score given a strong rebuttal to question 2.

**Limitations:**

The theoretical mismatch between approximated and true UCB sampling behavior is not acknowledged.

**Strengths And Weaknesses:**

## Strengths

**Soundness**

The recursive use of optimized posterior policies at every node is appealing and computationally grounded in convex optimization, with a clear KL‑regularized objective. The link between the optimized policy and equalization of UCB values provides a coherent interpretation that ties the algorithmic choice back to AlphaZero’s exploration principle. The breadth‑first, bottom‑up evaluation is internally consistent with the assumption that posterior policies can be computed locally using only $Q$ and $\pi_0$ at each node.

**Presentation**

The paper follows a logical progression (review of MCTS‑UCB → optimized posterior policy → RMCTS → intuitive example → experiments), which makes the narrative easy to follow.
The toy example (Section 5) concretely illustrates how recursion sets Q‑values and produces a strong posterior preference at the root, helping readers internalize the mechanics.

**Significance**

RMCTS directly targets a major hardware bottleneck (GPU latency in intra‑tree search) and demonstrates order‑of‑magnitude speedups via intra‑tree batching, which could materially affect practical planning systems. The approach is compatible with frameworks like MuZero, potentially broadening applicability beyond the specific games evaluated.

**Originality**

Pushing the Grill et al. optimized posterior policy from root‑only to all nodes in a recursive manner is a nontrivial conceptual extension. Recasting search as a dynamic‑programming‑like bottom‑up computation with intra‑tree batching is a fresh systems‑level contribution relative to standard sequential MCTS-UCB.

## Weaknesses

**Soundness**

RMCTS allocates $N(s,a)$ using $\pi_0(s,·)$ rather than visit counts that would arise from UCB, introducing a theoretical mismatch; the paper neither analyzes this bias nor provides an ablation to quantify its impact. Non‑adaptiveness means RMCTS predicts eventual UCB behavior from noisy Q’s and prior-biased $N(s,a)$; there is no robustness analysis to show this “fixed‑point shortcut” does not degrade performance relative to adaptive UCB in challenging regimes.

**Presentation**

Several definitions are unclear or appear late. E.g., the text clarifies in Section 4 rather than in Section 3 that Grill et al. originally suggested root‑only replacement, whereas this work applies it throughout the tree. Moreover, the definition of what a "simulation" entails is not specified. The paper also does not clearly explain how RMCTS plugs into self‑play training (e.g., which leaves are evaluated, how depths are chosen each iteration, and how variance/bias trade‑offs affect value targets).

**Significance**

Reported speedups emphasize single‑root scenarios, which favor RMCTS. In realistic inter‑tree training with many concurrent roots, MCTS‑UCB already amortizes latency; comparisons should better reflect this setting. In particular, why not evaluate MCTS-UCB with more root nodes than RMCTS to compare standard training regimes where the number of root nodes is not a bottleneck?

**Originality**

Although the recursive application is novel, many elements are extensions/combination of prior ideas (KL‑regularized policy improvement + MCTS), with relatively limited exploration of alternative divergences or variants beyond brief discussion.

---

> ### Author Rebuttal · Authors · 2026-03-31
>
> Thanks for all the points you have raised about the paper.
> We address your key questions below.
>
>
> Definition and interpretation of “simulations”:
> Every non-terminal node in the tree counts as one "simulation".
> This makes RMCTS align well with MCTS-UCB, which always
> ends a simulation when visiting a newly visited non-terminal node.
> Furthermore, if we visit a terminal node $k$ times, this is defined
> to consume $k$ "simulations". The total number of "simulations"
> in any branch is the total number of non-terminal nodes within it
> (included non-terminal leaves) as well as the total count
> on the terminal nodes in that branch. This coincides with
> the number of (true) simulations in an MCTS-UCB tree.
> We will clear this up in the camera-ready version.
>
>
> Theoretical mismatch between expected visit counts and
> true UCB visitation:
> There is indeed a mismatch in true visit counts,
> but the posterior policy is nevertheless quite
> close to the normalized visit counts of UCB.
> This was already demonstrated in the Grill paper, and
> we have observed this empirically even with relatively modest
> $N$ values (so long as $N$ is considerably larger than the action
> space.). For sufficiently large $N$,
> the posterior $\pi(a)$ is approximately
> $$\frac{\pi_0(a)}{ (\sqrt{N}/c) (Q(A^\ast) - Q(a)) + \pi_0(A^\ast)},$$
> where $A^\ast$ is the set of $Q$-maximal actions, and $\pi_0(A^\ast)$
> is the total probability mass on $A^\ast$.
> If we fix the $Q$ values, then iterating
> the procedure rapidly concentrates on $A^\ast$.
> This concentration on $A^\ast$ is immediately apparent from the asymptotic
> formula for large $N$. Crucially, however, due to the strict
> convexity of the divergence penalty, monotonic improvement and
> convergence hold for arbitrary $N$ as well, provided $\pi_0$ has full support.
>
> In practice however, when the $Q$ values are changing with
> each iteration, then there are issues with stability. It is
> perhaps best in the stochastic case to sample actions according to the current
> policy, but to maintain the divergence with the original $\pi_0$
> as an "anchor". We'd keep a running $N(a)$ count and
> $Q(a)$ estimate, and the divergence penalty, multiplied by $c/\sqrt{N}$,
> will slowly diminish. This is what we would do in the "multi-stage" variant.
>
>
> Integration with training loops:
> The result of RMCTS on any game state is a new value
> and policy, which should be an improvement over the original
> value and policy. We propose to use this policy as the "true"
> policy in training, the same as does MCTS-UCB.
> The value, on the other hand, also takes into
> account the final game score at the end of the rollout.
> We train the new value as a weighted combination of the
> RMCTS posterior value and final game score.
>
>
> Role of exploration when the network is fixed:
> Neural $Q$-values are inherently noisy, and maintaining the
> KL-divergence penalty acts as a necessary regularizer.
> In the literature it is standard to use MCTS in inference time
> policy computation as well as training.
> In this sense, we are merely proposing an alternative to MCTS
> (as we refer to RMCTS as a interchangable alternative in the paper) and show that in this
> context it has significant advantage. We are not attempting to make an argument about the
> larger class of policy iteration methods which severly alter or eschew MCTS for this.
>
>
> Fair comparison to MCTS-UCB in training settings:
> We selected 64 root states because that's the point where
> adding more states only produced marginal gains on our benchmarking hardware setup.
> Because the GPU was already fully saturated at 64 roots,
> increasing the number of concurrent states further would not
> meaningfully change the relative wall-clock speedup factors between
> the two algorithms.

---

> > ### Author Rebuttal · Reviewer_1Atp · 2026-04-01
> >
> > The authors have satisfactorily addressed my concerns. With the assumption that the additional clarifications will be incorporated into the main text of the paper, I have increased my assessment on the soundness and presentation, and improved my overall score.

---

### Official Review · Reviewer_7qEE · 2026-03-13

**Soundness:** 2
**Presentation:** 2
**Significance:** 2
**Originality:** 3
**Overall Recommendation:** 2
**Confidence:** 4

**Summary:**

This paper introduces Recursive Monte-Carlo Tree Search (RMCTS), a drop-in replacement for the standard AlphaZero MCTS-UCB algorithm. The key idea is to build the search tree in a non-adaptive, breadth-first manner guided by the prior neural network policy, and then recursively compute action values bottom-up using the regularized optimal posterior policy from Grill et al. (2020). At each node, the posterior policy is obtained by solving a local optimization problem (maximizing expected reward minus a KL-divergence penalty from the prior), which can be computed efficiently via Newton's method.

The main advantage claimed is massive parallelizability: since the tree is expanded breadth-first according to prior policies (not adaptively via UCB), all nodes at the same depth can be batched into a single GPU inference call. This amortizes GPU latency costs and yields speedups of 10-45x over MCTS-UCB for single-root-state search, and 2-14x for batched multi-root-state search. The authors demonstrate on Connect-4, Dots-and-Boxes, and Othello that while RMCTS is slightly weaker per-simulation than MCTS-UCB (due to its non-adaptive tree), the speed advantage more than compensates: RMCTS with 2x the simulations beats MCTS-UCB while still being 13x faster. Training time is reduced by roughly 3x.

The paper also discusses future directions including adaptive re-exploration variants and alternative divergence measures for the posterior policy optimization.

**Compliance With Llm Reviewing Policy:**

Affirmed.

**Key Questions For Authors:**

1. How does RMCTS perform on Go (9x9 or 19x19), a larger and more difficult task?

2. How does RMCTS compare to Gumbel MuZero in both speed and quality? Since Gumbel MuZero also uses regularized policy optimization for improved sample efficiency, and is the current standard in practice, this comparison is essential. Can you provide head-to-head results?

3. Can you provide formal convergence guarantees for RMCTS?

4. How sensitive is RMCTS to the quality of the prior policy? Since the tree expansion is entirely prior-guided, a weak prior (e.g., early in training) could lead to very poor tree structures. Does RMCTS degrade more than MCTS-UCB when the prior is weak?

**Limitations:**

yes

**Strengths And Weaknesses:**

**Strengths**
- Clear and elegant algorithmic design. The core idea of combining breadth-first prior-guided tree expansion with bottom-up recursive policy optimization is simple, principled, and well-motivated. The connection between the optimized posterior policy and equal UCB values (Section 3) is an insightful observation that provides good intuition.

- Clean algorithmic presentation. Algorithms 1-3 are precisely specified, the toy example in Section 5 effectively illustrates the recursion, and the proof of Newton's method convergence (Appendix B) is complete. The paper is generally well-written and easy to follow.

-  Code availability. An anonymous implementation is provided, which supports

**Weaknesses**
- Limited and narrow experimental evaluation.  The experiments are restricted to three relatively small board games (Connect-4, Dots-and-Boxes, Othello) using modest-sized ResNets (8 blocks, 48-64 channels). There are no experiments on more challenging domains like Go, Chess, or Shogi where AlphaZero is most impactful. It is unclear whether the favorable speed-quality tradeoff holds for larger games with deeper trees and larger action spaces. The paper would be significantly strengthened by experiments on at least one of these canonical benchmarks.
- No comparison with Gumbel MuZero and mctx. The paper positions itself against MCTS-UCB and mentions Gumbel MuZero (Danihelka et al., 2022) in the related work, but never compares against it experimentally. Since Gumbel MuZero is the current state-of-the-art for sample-efficient MCTS and also uses regularized policy optimization ideas, this is a critical missing baseline. The claim of addressing "a critical limitation in current state-of-the-art implementations" (Section 9) cannot be substantiated without this comparison.

- Missing theoretical analysis. Beyond the convergence proof for Newton's method, there is no formal analysis of RMCTS. Specifically: (a) no regret or convergence guarantees for RMCTS (unlike MCTS-UCB which has well-studied convergence properties); (b) no analysis of how the recursive value estimates compare to the true minimax values as N grows; (c) the claim that "playing strength becomes perfect as the number of simulations goes to infinity" (Section 7) for finite-state games is stated without proof.

---

> ### Author Rebuttal · Authors · 2026-03-31
>
> We would like to thank the reviewer for their thoughtful comments.
> Here are our responses to the questions raised.
>
> Gumbel MuZero and mctx:
> Regarding scaling to larger tasks like Go, and comparisons with Gumbel MuZero,
> we agree these are critical benchmarks. However, a rigorous empirical comparison
> requires a unified hardware and software framework (e.g., rewriting
> RMCTS in JAX to benchmark within mctx, or porting Gumbel MuZero to pytorch/C/TensorRT).
> While this extensive engineering integration is active future work, we
> conceptually contrast the two approaches here: Gumbel MuZero optimizes for
> sample efficiency but remains bottlenecked by sequential simulation paths.
> RMCTS optimizes for hardware efficiency via strict, non-adaptive
> breadth-first expansion, allowing for fully batched GPU inferences.
> Consequently, in identical wall-clock time, RMCTS evaluates a vastly larger tree.
> We will clarify this explicit trade-off between sample efficiency and
> hardware parallelization in the camera-ready version, emphasizing the
> decoupling of tree generation from the recursive policy update.
>
>
> Theoretical Analysis:
> Convergence guarantees:
> We can show that for large $N$, the posterior policy of RMCTS
> is asymptotically the same as for MCTS-UCB.
> In short, with sufficient simulations, the regularized optimized
> posterior policy always approaches the argmax of $Q$.
> But there is a more refined estimate concerning how large
> the posterior policy is at inferior actions.
> Let $A^\ast$ be the set of optimal actions
> where the true value $q(A^\ast)$ is maximal for the active player.
> Let $\pi_0(A^\ast)$ be the total prior probability mass on the optimal actions $A^\ast$.
> In general let $q(a)$ be the true value for any action $a$.
> The posterior $\pi(a)$ approaches
> $\pi_0(a) / ( (\sqrt{N}/c) (q(A^\ast) - q(a)) + \pi_0(A^\ast))$,
> meaning the ratio of $\pi(a)$ to this quantity approaches $1$ in
> the limit as $N \to \infty$.
> Because the denominator dominates for sub-optimal actions,
> where $q(A^\ast) - q(a)$ is positive,
> the regularized optimized posterior policy concentrates on $A^\ast$.
>
>
> Sketch of Proof that RMCTS approaches perfect minimax play as $N \to \infty$:
> Assume a finite state space and a prior policy with full support
> (i.e., $\pi_0(a) > 0$ for all legal actions). Because tree expansion is
> breadth-first and guided strictly by the prior, as the total simulation
> budget $N$ approaches infinity, the visitation count $N(s)$ for any node $s$ at any
> finite depth also approaches infinity.
> Because we set the KL-divergence penalty coefficient to
> $\lambda = c/\sqrt{N(s)}$, the punishment for diverging from the prior strictly
> drops to zero as $N(s)$ approaches infinity.
> Consequently, the local posterior policies at every node align
> perfectly with the unregularized greedy policy, concentrating entirely
> on the argmax of $Q$. During the bottom-up backup, the value estimate of a
> parent node is a weighted combination of the network's prior value estimate
> and the backed-up child values. As $N$ goes to infinity, the children hold an
> infinite "vote," completely overriding the network prior.
> Because terminal leaf states provide perfect ground-truth $Q$-values,
> this unregularized bottom-up backup exactly mirrors the standard minimax backup.
> By backward induction from the leaves, the root value and policy are guaranteed to
> converge to perfect minimax play. We will include this formal proof sketch in
> the appendix.
>
>
> Sensitivity to Prior Quality:
> The reviewer is correct that a purely non-adaptive breadth-first expansion is
> sensitive to a weak prior, potentially wasting simulation budget on unpromising
> branches. However, we have found empirically that during self-play training, the
> policy loss converges considerably faster than the value loss. Therefore,
> trusting the prior policy to allocate search budget is a highly reliable
> heuristic even relatively early in training. The algorithm can recover
> from a bad prior policy if the values discovered for non-favored
> actions is sufficiently high --- any large gap for example between an un-favored
> action and the current-best will immediately give a strong boost to the
> posterior policy.

---

> > ### Author Rebuttal · Reviewer_7qEE · 2026-04-04
> >
> > Thank you for the thoughtful rebuttal. The asymptotic convergence sketch — breadth-first expansion driving all visit counts to infinity, the KL penalty vanishing as λ = C/√N → 0, and backward induction recovering minimax play — is plausible and demonstrates correct intuition, as does the posterior concentration argument showing π̄ focuses on optimal actions. I appreciate the effort and would welcome a formalized version in the appendix.
> >
> > However, the two central experimental gaps remain unresolved. The paper claims to address "a critical limitation in current state-of-the-art implementations," yet the actual state-of-the-art (Gumbel MuZero) is never compared against. I understand the engineering challenge of cross-framework benchmarking (PyTorch/TensorRT vs. JAX/mctx), but the conceptual contrast between sample efficiency and hardware efficiency is precisely the hypothesis that requires empirical validation — we cannot assess whether RMCTS's speed advantage compensates for its sample efficiency loss relative to this stronger baseline without data. Similarly, the evaluation on only three small board games with modest networks leaves the scalability claims unvalidated for domains like Go where branching factor, tree depth, and prior accuracy pose fundamentally different challenges.
> >
> > On prior sensitivity, the claim that "policy loss converges faster than value loss" is the load-bearing argument for the non-adaptive expansion design, yet it is stated without learning curves or ablation data. On convergence theory, regret bounds and finite-sample analysis — the aspects most relevant in practice — remain unaddressed; only the asymptotic result is sketched. I am inclined to maintain my score.

---

> > > ### Author Response · Authors · 2026-04-08
> > >
> > > Thank you for the follow-up. We appreciate the opportunity to clarify the scope of our contribution and address your remaining concerns.
> > >
> > > Scope of claims: We agree that the phrasing in Section 9 may overstate our comparative claims. We will revise this text to scope our contribution directly to MCTS-UCB, rather than implying superiority over all state-of-the-art methods. Our core claim is that RMCTS introduces a new paradigm—non-adaptive breadth-first expansion combined with recursive bottom-up policy optimization—that fundamentally unlocks intra-tree parallelism. This is orthogonal to Gumbel MuZero's focus on sample efficiency. A direct empirical comparison requires substantial cross-framework engineering that we believe is best served by a dedicated follow-up study.
> > >
> > > Prior sensitivity and learning curves: We have the learning curves from our training runs demonstrating that the policy loss converges substantially faster than the value loss across our test environments. We will include these plots in the revised appendix to empirically support our reliance on the prior for guiding the non-adaptive expansion.
> > >
> > > Scalability: We note that Grill et al. (2020), on which our posterior policy optimization is based, similarly validated their initial approach on smaller environments. We believe the foundational algorithmic insight, that the regularized posterior can be computed locally and composed recursively, is a general contribution that already merits presentation to the community now, prior to scaling to 19x19 Go, which requires significant additional compute and infrastructure.
> > >
> > > Finite-sample analysis: We acknowledge this is an important open question. The asymptotic result we sketched establishes correctness; deriving tighter finite-sample regret bounds for this specific recursive optimization would require new proof techniques that we view as a substantial theoretical contribution in its own right, suited for future work.

---

### Official Review · Reviewer_ecQ2 · 2026-03-16

**Soundness:** 3
**Presentation:** 3
**Significance:** 3
**Originality:** 3
**Overall Recommendation:** 4
**Confidence:** 3

**Summary:**

The authors propose a Monte Carlo tree search variant that decouples tree expansion from value backup. Tree generation follows the prior policy only, so all nodes at the same depth can be expanded in parallel. Value estimation is then performed bottom-up, from the leaves to the root. At each state, the algorithm computes a KL-regularized posterior policy from the prior policy and the local action-value estimates. This localization of policy improvement relies on the observation of Grill et al. (2020) that the regularized posterior can be computed locally from Q(s, ·) and π0(s, ·).

Unlike AlphaZero’s MCTS-UCB, where the posterior policy is formed only after sequential simulations and is read off from normalized visit counts, RMCTS explores the tree in parallel according to the prior and then applies KL-regularized policy improvement recursively. Both the posterior policies and the action-value estimates are computed bottom-up through recursive backup.

Empirically, the authors show that RMCTS matches or exceeds the quality of MCTS-UCB while using much less wall-clock time. They report that RMCTS-trained networks reach comparable quality in roughly one-third of the training time on Othello, Dots-and-Boxes, and Connect-4.

**Compliance With Llm Reviewing Policy:**

Affirmed.

**Key Questions For Authors:**

* What other training environments would be viable to consider training on? There are small variants of chess with Stockfish analogues  that could be used for small chess games. Checkers or backgammon could be  another choice in case you are constrained by compute.

* How would you adapt your method in a MuZero style of exploration in the latent space?

**Limitations:**

yes

**Strengths And Weaknesses:**

Strengths:
* The paper offers a very compelling method that is able to parallelize training in games. Non-parallelizability is a grave obstacle to scaling these game-solving algorithms.
* The authors were able to use a simple intuitive theoretical observation to design a practical algorithm with very favorable properties.
* The preliminary experiments of the paper spell out the supremacy of the method.

Weaknesses:
* The experiments are rather limited. I imagine that an experiment even in a small Chess or Go variant might be prohibitive without compute budget; still, the authors could consider different standard RL experiments.

* The theoretical contribution is limited and the Newton method's convergence proof has no rates.

---

> ### Author Rebuttal · Authors · 2026-03-31
>
> We thank the reviewer for their feedback on the paper.
> We address your specific questions regarding scaling and convergence below.
>
> What other training environments would be viable to test on?:
> We agree that small variants of chess or checkers are excellent
> intermediate benchmarks. As discussed in our response to
> Reviewer EUZM regarding "Scope and Scaling," our immediate focus for
> this initial paper was establishing the fundamental hardware-efficiency
> and theoretical convergence of RMCTS on environments where we could
> fully study the algorithmic mechanics. Moving forward, utilizing Stockfish
> analogues for small chess variants is exactly the type of environment we
> plan to target next to test RMCTS's capacity for deep tactical search.
>
> How would you adapt your method to a MuZero framework?:
> RMCTS seamlessly integrates into the MuZero framework.
> Standard MuZero utilizes MCTS-UCB over a learned latent space;
> we can simply replace this with RMCTS. The exact environment simulator
> is replaced by the learned dynamics model $g(s, a) = (r, s')$, and the
> prior policy and value are provided by the prediction function $f(s) = (\pi_0, v)$.
> RMCTS's breadth-first expansion simply operates over these latent states and
> intermediate rewards. Because RMCTS drastically increases the tree size per
> GPU second through batching, it acts as a much faster, fully-batched policy
> improvement operator for the MuZero target network.
> (Note: Comparisons with Gumbel MuZero are discussed in our response to 7qEE).
>
> Convergence rate of Newton's method:
> In our inner optimization loop, Newton's method is guaranteed to converge quadratically.
> We are finding the root $u$ of a function $f(x)$ that is strictly decreasing and
> strictly convex. Because our initial guess is strictly to the left of $u$, the sequence
> monotonically increases, limiting on $u$ extremely rapidly. Specifically, the
> asymptotic error ratio follows:
> $$\lim_{n \to \infty} \frac{e_{n+1}}{e_n^2} = - \frac{f''(u)}{2 f'(u)}.$$
> Because of this strict quadratic convergence,
> it typically requires only a handful of iterations to reach floating-point precision.
> We will add a formal reference for this to the appendix.
>
> In the table below we show round robin scores in Connect 4 between
> 12 different networks.  Networks m1 - m6 were trained using MCTS-UCB.
> Networks r1 - r6 were trained by RMCTS.  In both cases we trained with a batch of 64 simultaneous root states
> to mitigate GPU latency for both methods.
> The index coincides with the
> number of hours spent training it.   So for example m5 is the MCTS-UCB network
> trained in 5 hours.   Each score is the accumulated scores of 100 games, with 50 games as player 1, and 50 games as player 2.   In each case, the raw network is supplemented by
> MCTS-UCB using 64 simulations.   Each game begins at the initial state,
> and a cold temperature of 0.2 was used to select actions.  The main point we want
> to make here is that this table shows the superiority of RMCTS training over MCTS-UCB.
> In particular, m3 loses to r1, and m6 loses to r3, backing our claim of a 3x speedup
> in training time.   We can do a similar table for the other two games, and add this information
> to the appendix.
>
> | | m1 | m2 | m3 | m4 | m5 | m6 | r1 | r2 | r3 | r4 | r5 | r6 |
> | :--- | :--- | :--- | :--- | :--- | :--- | :--- | :--- | :--- | :--- | :--- | :--- | :--- |
> | **m1** | 0 | -39 | -67 | -74 | -77 | -74 | -62 | -70 | -83 | -93 | -90 | -97 |
> | **m2** | 39 | 0 | -38 | -31 | -51 | -67 | -7 | -69 | -79 | -82 | -94 | -88 |
> | **m3** | 67 | 38 | 0 | -20 | -6 | -40 | **-8** | -51 | -75 | -90 | -87 | -91 |
> | **m4** | 74 | 31 | 20 | 0 | -9 | -17 | 2 | -38 | -59 | -83 | -82 | -83 |
> | **m5** | 77 | 51 | 6 | 9 | 0 | -6 | -7 | -38 | -58 | -79 | -82 | -87 |
> | **m6** | 74 | 67 | 40 | 17 | 6 | 0 | 16 | **-44** | -62 | -77 | -76 | -83 |
> | **r1** | 62 | 7 | **8** | -2 | 7 | -16 | 0 | -69 | -77 | -96 | -88 | -96 |
> | **r2** | 70 | 69 | 51 | 38 | 38 | **44** | 69 | 0 | -51 | -66 | -71 | -81 |
> | **r3** | 83 | 79 | 75 | 59 | 58 | 62 | 77 | 51 | 0 | -38 | -50 | -61 |
> | **r4** | 93 | 82 | 90 | 83 | 79 | 77 | 96 | 66 | 38 | 0 | -8 | -22 |
> | **r5** | 90 | 94 | 87 | 82 | 82 | 76 | 88 | 71 | 50 | 8 | 0 | -5 |
> | **r6** | 97 | 88 | 91 | 83 | 87 | 83 | 96 | 81 | 61 | 22 | 5 | 0 |

---

> > ### Author Rebuttal · Reviewer_ecQ2 · 2026-04-06
> >
> > My concerns are resolved. Yet, due to the limited experimental results I decided to keep my score.

---

### Decision · Program_Chairs · 2026-04-30

**Decision:**

Accept (regular)

**Comment:**

This paper considers an alternative objective function on Monte-Carlo Tree Search that recursively applies the reguralization-based weights. The search part follows standard policy $\pi_0$ whereas the Q value of the policy is recursively calculated (RMCTS, Algorithm 1) and efficiently implemented. Newton's method on the analysis of optimal value is provided. The benefit of recursive computation is shown in a stylized example in Section 5. RMCTS empirically outperforms the closest baseline of Grill et al. 2020.

Let me summarize the novelty of the work in the followings:

A Vanilla version of Bandit-based MCTS (UCT) was proposed around 2006 (Kocsis and Szepesvari 2006, Coulom 2007) and at that time it explores based on UCB values (based on rewards or "value function"). The naive version of UCT does not perform well and there are many heuristics encoded in practical settings like Game of Go. AlphaGo (around 2015) relies on a policy function that estimates the next move based on its probability to be chosen. The use of policy function is crucial because the size of the tree is exponentially large to the depth and pruning the tree makes the search exponentially efficient. Since then, people consider MCTS as a combination of policy function and value function (lets say policy-guided UCT).


Grill et al. 2020 showed that the regularized optimization (p3 left, below "maximizing the objective function…" in this paper) is the "equilibrium" of the policy-guided UCT. Therefore, running the reguralized UCT asymptotically matches the policy-guided UCT and its performance is better in finite time.

That said these are key issues remain: (A) these policy-guided UCT does not scale. Another issue is that (B) the behavior of reguralized UCT is too much dependent on policy (Section 5 "A simple example"). Even if it finds some good action sequence based on value function, it can be ignored if the policy's weight is low. What this paper proposes is that first, do breadth-first search by policy and from the bottom, recursively recompute the value based by recursively applying the policy-reguralized value function. Regarding (A), since it is based on predefined policy, it is easily parallelizable. Regarding (B), it can weigh the best action (action of largest value) enough if the $\pi(A_{best})$ is not too small compared with $\pi(A_{other})$ of the other action on the same branch. So, among these policy-guided UCT variants this algorithm looks the most reasonable one to me. For one of the most important topic in AI (MCTS), two major issues are addressed in this paper. Therefore, I think this paper has a potentially high impact.

Limitations: are that limited scale of empirical evaluation (Reviewer ecQ2, EUZM), lack of comparison with Gumbel MuZero (Reviewer 7qEE), Inter-tree extension (Reviewer 1Atp), lack of adaptivity (Reviewer EUZM). Reviewers generally like the paper. Reviewer 7qEE is not fully convinced, but Given that Gumbel MuZero is from an industry lab and comparison with it is not a mandatory requirement. Another possible criticism is that the paper's description (depth-first search in Algorithm 1) is not faithful and does not describe what is actually implemented.

Summary: A conceptually simple modification that many reviewers are interested and written in a very minimal and straightforward way, which implies a large impact. I would endorse the paper in the aforementioned ground.